

# Uncertainty in flood frequency analysis of hydrodynamic model simulations

Xudong Zhou[1], Wenchao Ma[1], Wataru Echizenya[2], and Dai Yamazaki[1]

[1]Institute of Industrial Science, The University of Tokyo, 4-6-1, Komaba, Meguro-ku, Tokyo, 153-8505, Japan
[2]Corporate Planning Department, MS&AD InterRisk Research & Consulting, Inc., 2-105, Kanda Awajicho, Chiyoda-ku, Tokyo 101-0063, Japan

**Correspondence:** Xudong Zhou: x.zhou@rainbow.iis.u-tokyo.ac.jp

**Abstract.**

Assessing the risk of a historical-level flood at a large scale is essential for regional flood protection and resilience establishment. Due to limitations on the spatiotemporal coverage of observations, the risk assessment relies on model simulations thus is subject to uncertainties from various physical processes in the chain of the flood frequency analysis (FFA). This study

assessed the FFA performance as well as the uncertainties with different combinations of FFA variables (river water depth and water storage), fitting distributions and runoff inputs based on the flood characteristics estimated by a global hydrodynamic model CaMa-Flood. Results show that fitting performance is better if FFA is conducted on river water depth and if Wakeby function is selected as the fitting distribution. Deviations in the runoff inputs are the main source of the uncertainties in the estimated flooded water depth based on point analysis. This deviation is relevant to the model ability to reproduce the mean

state of annual maximum flood extent and it is almost homogeneous for different flood return period. The uncertainty resulted from fitting distributions increases from the regular period to the rarer floods. The regional investigation of high-resolution inundation area over the lower Mekong River basin shows similar statistics as the point analysis, implying a large uncertainty with 20% deviation in the total inundation area between different runoff inputs. Regional validation of the CaMa-Flood with two other flood hazard maps proves the reliability of the inundation in space and values. Global analysis on the floodplain

water depth implies an increasing contribution of uncertainties in fitting distribution to the total uncertainties for rarer floods in almost all land grids. While the changes in contribution of uncertainties in runoff inputs differentiates in regions. The much higher contribution of runoff uncertainty for rarer floods in wet/flat regions necessitates special attention on rainfall-runoff model calibration (or runoff bias correction) if gauge discharge observations are available. Different adaptions to the large floods are needed for regions with different flood water depth and with different inundation agreements among simulations.

.



# 1   Introduction

Flood frequency analysis (FFA) is one way of finding the occurrence and identification of large floods based on limited length of dataset (Hamed and Rao, 2019). It is of vital importance for the analysis of flood control and design of many mitigation projects. The result of FFA can be also used in flood risk assessment which is helpful for stakeholders and insurance services.

FFA was introduced more than 30 years ago (Liscum and Massey, 1980; Wiltshire, 1986), and has been applied to multiple regions in different continents and climates where gauge observations of river status (e.g., river discharge, river water depth) are available.

Because most of the gauge observations have been collected for periods of time significantly less than 100 years in GRDC (Global Runoff Data Centre), the estimation of the "design discharge" (design stage, or water level at a high flood-frequency)

necessitates a degree of extrapolation based on curve-fitting to the existing data (Kidson and Richards, 2005). The limitation of FFA is therefore apparent as the fitting requires a priori assumption about the underlying distribution generating flood events. Though, because limited length of observational records cannot represent the complete characteristics of floods, a range of more-or-less skewed, relatively complex distributions is always together considered to account for the uncertainties. Typical distributions that are used include Pearson III type, Log-Pearson, Gauss, Gumbel and Log-normal distribution (Radevski and

Gorin, 2017; Drissia et al., 2019). However, no conclusion is found whether any of the fitting distributions is preferable for most of the regions (Drissia et al., 2019). Different distributions are recommended to test with local records before selecting the one with the best performance.

FFA is generally performed for gauge records, while there are many data-poor regions or ungauged regions that also suffering floods disasters, such as the Indus floods in Pakistan (2010) and recent floods in Sri Lanka (2020 May). The characterization of

flood behaviour in data-poor regions has received considerable attention (Blazkova and Beven, 2002; Bernhofen et al., 2018) while the flood estimations in these ungauged catchments are challenging without enough records (Salinas et al., 2013; Trigg et al., 2016). There are two families of solutions emerging to facilitate discharge estimation in data-poor regions (Smith et al., 2015). The first is to relate flood-frequency behaviour in similar catchments with observational records. The second comprises rainfall-driven model cascades that attempt to estimate the river flow through hydrological processes, which is regarded as

"continuous simulation" (Bras et al., 1985; Beven and Hall, 2014).

Continuous simulation is defined as a methodology being developed for estimating flood frequencies where no flood records exist (Kjeldsen et al., 2014). The results of FFA based on continuous simulations are therefore subject to uncertainties propagated from the rainfall, rainfall-runoff models and the routing process which routes the generated runoff to river flow at river profiles of interest (Trigg et al., 2016; Bernhofen et al., 2018). Together with the uncertainties resulting from the fitting dis-

tribution, the flow magnitude at the specific "design discharge" will be uncertain. The uncertainties in the FFA are calculated to be the most important source of uncertainty in flood risk assessments which relate the inundation estimations (Merz and Thieken, 2009).

Associating the FFA analysis and inundation area provides the way to evaluate the potential flood damages for a given magnitude of flood. This first requires a method to estimate the inundation area and links it to the water level. Qi et al. (2009)



connected the Poyang Lake area extracted from Landsat images and *in-situ* water level measurements. The relation was then extrapolated to obtain the inundation area according to the frequency of water level. Though, this is only valid for large open water bodies rather for floodplains where inundation is not frequent. Alternatively, the inundation area can be estimated by statistical models (Sarhadi et al., 2012; Odunuga and Raji, 2014) or physical-based models Merwade et al. (2008) which relate

the inundation area with calculated floods. The global hydrodynamic river model CaMa-Flood (Catchment-based Macro-scale Floodplain, Yamazaki et al., 2011, 2012) is able to route estimated runoff from various rainfall-runoff models to provide the estimates of flow characteristics (e.g., discharge, water level, water storage in a river channel or floodplain) at all the model points. The inundation area corresponding to a given level of flood (e.g., 100 years return period) can be achieved by downscaling the FFA results to high-resolution maps with bias-corrected topography data.

FFA can be conducted on any characteristics of river flow, but mainly with river discharge and water stage (or named water level or similarly water depth) because they are normally recorded as gauge observations (Radevski and Gorin, 2017). There is no preference of these two variables and the selection is determined only by the data accessibility. The results of FFA based on the discharge will slightly different from the results with the water stage because of the loop rating curve relationship between discharge and water stage (Domeneghetti et al., 2012; Alvisi and Franchini, 2013). In addition to the aforementioned

uncertainty sources from rainfall, rainfall-runoff model, routing processes, fitting and downscaling, the uncertainties of the inundation area corresponding to a certain level of floods will be complex and remain un-investigated.

In this study, Flood Frequency Analysis is applied to the flow estimation by CaMa-Flood and the resulting uncertainties are assessed at various spatial scales on different flood characteristics. Methodologies are introduced in section 2. In section 3, the performance of FFA is compared among the flow variables used for FFA, the fitting distributions as well as the runoff that

drives CaMa-Flood. Uncertainties resulted from different sources are quantified in the Mekong deltas, with the spatial characteristics shown and agreement of different settings over the inundation estimation evaluated in section 4. A global overview of the uncertainties in floodplain water depth and the contribution from sources is provided in section 5. The discussions and conclusions are followed in the end.

## 2 Methodologies

### 2.1 CaMa-Flood

The CaMa-Flood (Catchment-based Macro-scale Floodplain) model is designed to simulate the hydrodynamics in continental-scale rivers. The entire river networks are discretized to irregular unit-catchments with the sub-grid topographic parameters of the river channel and floodplains. The river discharge and other flow characteristics can be calculated with the local inertial equations along the river network map. Water storage of each unit-catchment is the only prognostic variable that to be solved

with the water balance equation. The water level and flooded area are diagnosed from the water storage at each unit-catchment using the sub-grid topographic information. Detailed descriptions of the CaMa-Flood can be referred to the original papers by Yamazaki et al. (2011, 2012, 2014).





The major advantage of the CaMa-Flood simulations is the explicit representation of flood stage (water level and flooded area) in addition to river discharge. This facilitates the comparison of model results with satellite observations, either the altimeters by SAR or inundation images by optical or microwave imagers. The estimation of the flooded area is helpful for assessment of flood risk and flood damages by overlaying it with socio-economic datasets.

Another apparent advantage of the CaMa-Flood is its high computational efficiency of the global river simulations. The CaMa-Flood utilizes a diagnostic scheme at the scale of unit-catchment to approximate the complex floodplain inundation processes. The prognostic computation for water storage is optimized by implementing the local inertial equation and the adaptive time step scheme. The high computational efficiency is beneficial for implementations at a global scale. This is critically important as ensemble simulations are frequently applied to account for uncertainties but computation time will be

multiplied manyfold. In this study, the CaMa-Flood was driven by the various runoff inputs to achieve the flow characteristics at each unit-catchment at the global scale.

## 2.2  Experiments design

The uncertainties to be investigated in this study are attributed to three major sources as (1) the variables used for the FFA, (2) the fitting distributions used for FFA and (3) the runoff inputs to the CaMa-Flood. For the variables selection, V1_($rivdph$)

represents the FFA is based on the numeric results of "river water depth" provided by CaMa-Flood. In V2_($sto2dph$), the FFA was first conducted on the estimated water storage which is the only prognostic variable in the CaMa-Flood. Then at each return period (e.g., 100 yrs, 50 yrs), the river water depth was estimated based on the storage-water depth relation and the corresponding water storage. Because of the non-linear relation between water level and storage, the fitting will lead to different results. The differences between experiment V1 and V2 denote the uncertainty resulted from the selection of target

variables used for FFA.

The uncertainty due to the fitting distributions used for FFA was evaluated as the resulting differences by applying various fitting functions (i.e., F1 – F6). These distributions are generally used in FFA but for different variables in different fields, and they were treated without priorities in this study. The samples were fitted automatically without any manual modifications in their parameters with L-moments optimization.

The results of FFA were based on the output of CaMa-Flood which is associated with the runoff inputs. In this case, the CaMa-Flood were driven by seven different kinds of runoff forcing (i.e., R1 – R7) from eartH2Observe (e2o) category (Schellekens et al., 2016). The runoff were driven by the same WFDEI (WATCH Forcing Data methodology applied to ERA-Interim data, Weedon et al., 2014) but with different land surface and hydrological models, therefore, the runoff inputs have already contained the uncertainties in the forcing and that in the rainfall-runoff model processes (model structures and model

parameters). The deviation of the results in the FFA among the seven inputs was, therefore, the uncertainty caused by runoff inputs.





**Table 1.** Various experiments used in this study for uncertainty analysis. There are three groups as (A) the variables for FFA (B) the fitting distributions and (C) the runoff inputs. Different runoff are generated by same forcing (WFDEI) but different land surface models or global hydrological models (specified in the bracket).

| A | Variables | B | fitting distribution | C | Runoff |
|---|---|---|---|---|---|
| V1 | $rivdph$ | F1 | GEV (Generalized Extreme Value) | R1 | e2o_anu (W3) |
| V2 | $sto2dph$ | F2 | GAM (Gamma) | R2 | e2o_cnrs (ORCHIDEE) |
| | | F3 | PE3 (Pearson III) | R3 | e2o_jrc (Lisflood) |
| | | F4 | GUM (Gumbel) | R4 | e2o_ecmwf (HTESSEL) |
| | | F5 | WEI (Weibull) | R5 | e2o_nerc (JULES) |
| | | F6 | WAK (Wakeby) | R6 | e2o_univk (WaterGAP3) |
| | | | | R7 | e2o_univu (PCR-GLOWB) |

## 2.3 Flood frequency analysis (FFA)

The runoff inputs are available from 1980 to 2014 (35 years in total) with a spatial resolution of 0.25º (∼25km at the equator). For a specific unit-catchment defined in the CaMa-Flood, the maximum value of the daily river water depth or catchment storage was obtained for each year and sorted. The frequency as the return period ($P_m$) was calculated with the following
equation:

$$P_m = \frac{m}{N+1},\tag{1}$$

where $m$ is the sorted ranking, $N$ denotes the number of total years (herein 35).

Then the parameters of the fitting distributions were calculated with the basis on these sorted annual values with the L-moment method (Hosking, 2015; Drissia et al., 2019). It is defined as a linear combination of probability-weighted moments of the time series. The parameters estimation using L-moment and quantile functions used for different distributions have been
described in detail in Hosking (1990). The computation of the parameters was done in Python *lmoments3* Library. Note that only the Wakeby (WAK) is a 5-parameters function while the others are all 3-parameters functions.

## 2.4 Criterion

Akaike Information Criterion (AIC, Sakamoto et al., 1986; Mutua, 1994) was used to evaluate the performance of the FFA
against the annual values. $aic$ is calculation as

$$aic = 2k + n \cdot log(\frac{\sum (S-O)^2}{n}),\tag{2}$$

where $k$ is the number of parameters needed for the fitting distribution, $S$ represents the simulated values, $O$ represents the observed values, $n$ denotes the number of samples. The performance of fitting is better when the $aic$ value is lower because of smaller deviations between simulations and observations.



## 2.5 Study area and downscaling to high-resolution inundation map

To reduce the computation cost due to high-resolution simulations, the CaMa-Flood was run globally at a 0.25° spatial resolution, which means only one unit-catchment was assigned for each 25 km by 25 km grid. The evaluation of the FFA performance with $aic$ was conducted at the global scale to capture the overall features, corresponding to the results in section 3.

It is difficult to characterise the river water depth or inundation area in detail with local topography at a low resolution (0.25°), and it is difficult to visualize the inundation map at a high resolution (<100m) for the globe. Therefore, high-resolution (3 arcsec, ~90 m at the equator) regional analysis related to the floodplain water depth and inundation area with their uncertainties was conducted regionally over the lower Mekong River basin, where the delta is vulnerable to floods (Shin et al., 2020). Corresponding results from point analysis to regional analysis on the uncertainties in water depth and inundation area will be

presented in section 4.

The estimated low-resolution storage was downscaled to the high-resolution inundation map with the topography map MERIT (Multi-Error-Removed Improved-Terrain DEM, Yamazaki et al., 2017) at 90 m. The fundamental assumption is that the water surface is flat within each unit-catchment and the total water storage under the identical water level should be equal to the water storage estimated in this unit-catchment (see Figure 1-a). The relationship between the water level and water

storage or the flooded area should be similar to Figure 1-b, as when the floodplain has been inundated, the small increases of water level is corresponding to large changes in the water storage as well as the flooded area. River water depth can be saturated after inundation (it does not react significantly to the increase of storage after flooding), and this might cause the error in function fitting. The assumption of the flat water surface is not valid for long river sections or large water bodies (e.g., large lakes or reservoirs with water surface gradient) and rivers with large slopes (e.g., mountainous area). However, the impact of

violation is limited at the catchment scale with a grid size of 25 km (in consistency with the global scale). Inundation area over the mountainous area is also limited compared to that in the floodplains.

## 3   Comparisons among different experiments

In this section, the performance of the FFA will be inter-compared among different experiments given in Table 1. The lower $aic$ indicates the better fitting performance. Comparisons are conducted in three groups. In the first *variables* group, the two

variables on which the FFA was based are compared. In the second group, the fitting performance is compared among using different fitting distribution. In the last group, the FFA performance determined by different runoff inputs is compared.

### 3.1   Comparison between using different *Variables*

Water storage is the prognostic variable in the CaMa-Flood that transfers water from the upstream to the downstream. It is estimated by the inflow to this catchment, added runoff within the catchment and outflow to the downstream catchment.

Whereas, river water depth is co-determined by the river water storage, river channel cross-section and floodplain topography profile. Estimating the river water depth hence includes extra information as well as the uncertainty from the topography. This

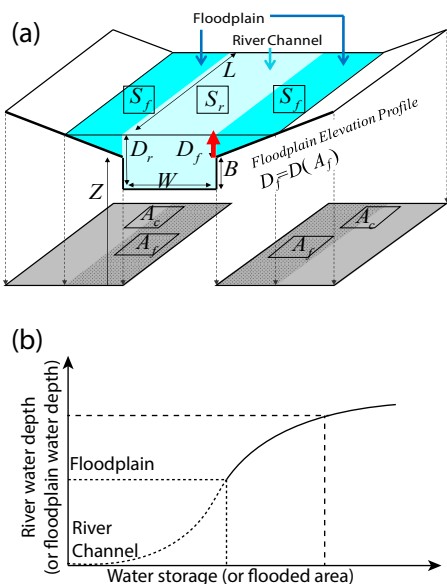

**Figure 1.** (a) Illustration of a river channel reservoir and a floodplain reservoir defined in each unit-catchment. The water level for the river channel and floodplain is assumed to be the same in each unit-catchment. The denotation of each parameters and its calculation can refer to (Yamazaki et al., 2011). (b) The relationship between the water level and water storage as well as the flooded area for a specific unit-catchment. The shape of the curve within the river channel is determined by the profile of the river channel and the curve above the river channel is mainly affected by floodplain topography.

section evaluates how the fitting distributions work for the two different variables. Because the river water depth and the river water storage are not in the same unit or same magnitude, they were normalized to the range of [0,1] for each grid divided by the maximum value for each unit-catchment. The fitting distributions (i.e., F1 – F6, Table 1) were applied to fit the modelled time series. The fitting performance was evaluated by the $aic$ value. The estimated $aic$ values for the two variables were compared

and one sample for e2o_ecmwf and GEV function is shown in Figure 2.

Because the time series were normalized to ranges of 0 and 1, the fitting performance is relatively high with low $aic$ (<-50) in most of the unit-catchments. Low fitting performance is found in the Greenland area and those dry areas in the Sahara, Mongolia and middle Australia (Figure 2-a). The area with low fitting performance (high $aic$) increases when dealing with the storage, typically in Mongolia, Australia, South Africa, south Latin America and in the west part of North America. These

10 regions are mainly dominated by dry climate or mountainous topography. The relatively low river discharge could be the reason for low model performance in the fitting.

The difference of the $aic$ values for the river water depth and that for the storage is mapped as Figure 2-c. In which, negative values indicate that the fitting performance is better for water depth than for that for the water storage. Despite the near-zero values, negative values (red scatters) are distributed in the main parts of the world. The places with the largest differences are

15 distributed in the northern and southern Africa, Australia, Northern China, Western America, in high consistency with the high

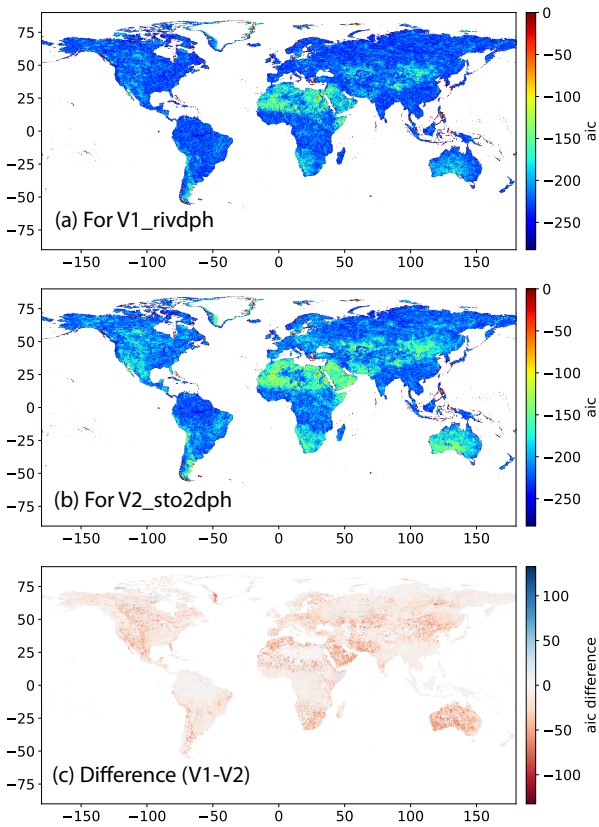

**Figure 2.** Performance of flood frequency analysis for (a) V1_$rivdph$ and (b) V2_$sto2dph$. The performance was quantified with $aic$ and (c) is the $aic$ difference of (a) and (b). Negative difference indicates better performance of FFA for V1_$rivdph$. This is only an example for e2o_ecmwf and GEV fitting distribution.

values in Figure 2-b. Although positive values are also found, the values are not large. The results indicate that for most of the lands, the fitting on the data of river water depth is better than the fitting on the water storage. Though this is only the results of a case with e2o_ecmwf runoff input and GEV distribution.

An overall evaluation on all distributions and runoff inputs are shown in Figure 3. The probability distribution of the $aic$ values for all the global grids are plotted in Figure 3-a and 3-b for V1_$rivdph$ and V2_$sto2dph$, respectively. We found that the fitting distribution determines the $aic$ values as the pdf curves for the same distribution always gather together. This is more distinguishable for the water depth than that for the water storage. The pdf curves have two peaks, one is normally distributed with mean values around -200 (or -220) and the other one is near zero. The later peak corresponds to the red scatters in Figure 2-a,b, showing poor fitting performance of the distributions over the coastal regions. Regarding the differences among different distributions, WAK (yellow lines) have the smallest $aic$ values with the best performance while GAM (red lines) and GUM (black lines) have the largest values with the poorest performance in Figure 3-a. The other three distributions (GEV,





PE3 and WEI) have a similar and moderate performance for the water depth. Although the lines were not so distinguishable in Figure 3-b, the sequence of the fitting performance for different distributions is the same as for the water depth.

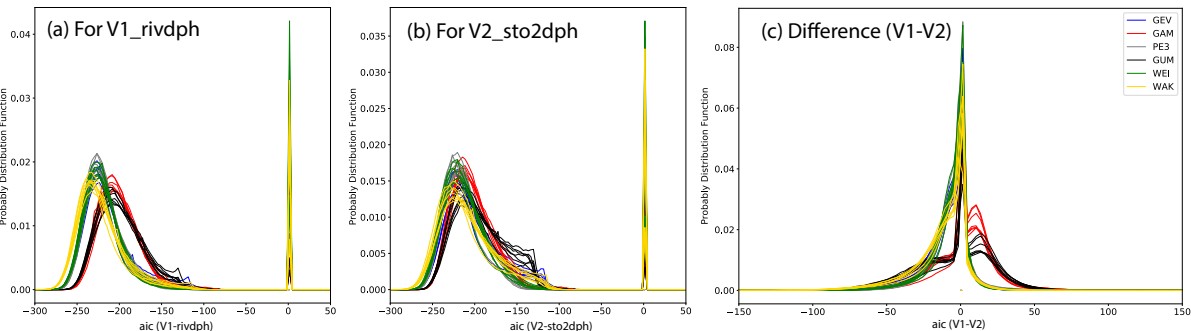

**Figure 3.** Overall performance of flood frequency analysis for (a) V1_$rivdph$ and (b) V2_$sto2dph$. The performance $aic$ over all the land grids are collected and displayed as the histogram. (c) is the $aic$ difference of (a) and (b). Negative difference indicates better performance of FFA for V1_$rivdph$. Different colors represent different fitting distributions, and the multiple lines in a specific color represents results driven by different runoff inputs. The type of the runoff inputs are not specified in these three graphics.

Figure 3-c shows the difference of fitting performance for water depth and water storage (corresponding to Figure 2-c if e2o_ecmwf and GEV is specified). Same as Figure 2-c, negative values indicate that the fitting performance for water depth

is better than that for the water storage. For the distributions of WAK, GEV, PE3 and WEI, more negative values were found especially within the range of [-50, 0]. While for GAM and GUM, more positive values are found within the range of [0, 25], showing better performance for water storage than that for water depth. But as we see from Figure 3-a and 3-b, the fitting performance of GAM and GUM is not as good as other functions. We, therefore, can conclude that the fitting is better applied to the water depth (V1_$rivdph$) rather than the water storage (V2_$sto2dph$). Since the normalization did not change the relative

magnitude of different values, the difference between fitting river water depth and water storage results from their relationship (Figure 1). For the floods (tails of the fitting distribution), the changes in water storage should be larger than that changes in the water level if given a shift of the flood frequency. This causes the resulting difference in the fitting performance.

### 3.2 Comparison between different *Fitting distributions*

In order to find the better fitting distribution for FFA, we ranked the fitting performance by in distributions according to the $aic$

values at each unit-catchment. The distribution with the best performance (the smallest $aic$ value) was scored 6 and the function with the worst performance (the largest $aic$ value) was scored 1. The other distributions were scored from 2 to 5 according to the sequences of $aic$ values. The scoring results for runoff e2o_ecmwf with V1_$rivdph$ are discussed in this subsection as an example (Figure 4).

The WAK is scored as the best function in most of the global grids (Figure 4-a) except the dry areas in Sahara, Central

Asia and middle southern Australia. This is mainly because WAK is a 5-parameters function and might be overfitted while all


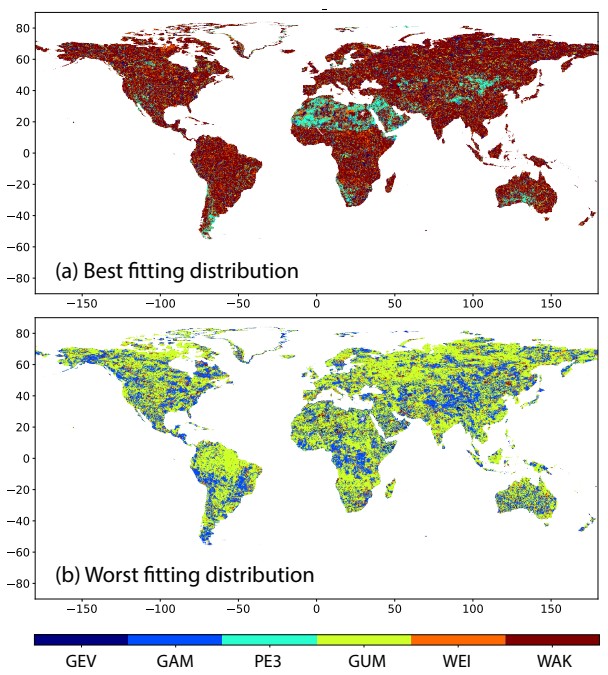

**Figure 4.** The type of fitting distribution corresponding to (a) the top ranking and (b) the lowest ranking of the FFA performance according to the $aic$ criterion.

other functions only have three parameters. Despite the best distribution, the PE3, GEV and WEI are marked as the second-best functions in different parts of the globe. While the GAM and GUM are generally ranked the last implying that the two functions are not suitable for FFA on the water depth (Figure 4-b). The same results have been shown in Figure 3, as $aic$ for GAM and GUM are in a large probability higher than $aic$ with other fitting distributions.

5    The average score for all the global catchments shows the same results (Figure 5). No matter the average water depth ($rivdph$) is, the WAK ranks the first. WAK, PE3 and WEI have similar performance when the river water depth is less than 1.0m, corresponding to the land grids where this is a low probability to suffer heavy floods. For the grids with average river water depth larger than 1.0m, the performance becomes more distinguishable as WAK outperforms other functions but the differences among GEV, PE3 and WEI become small. The GAM and GUM always have the worst performance for all ranges 10    of the water depth.

The same comparison is not conducted for other runoff inputs or V2_$sto2dph$ because from Figure 3 we can conclude that the performance is mainly determined by the fitting distribution. The differences between different runoff will not change the fitting performance, so the ranking scores.

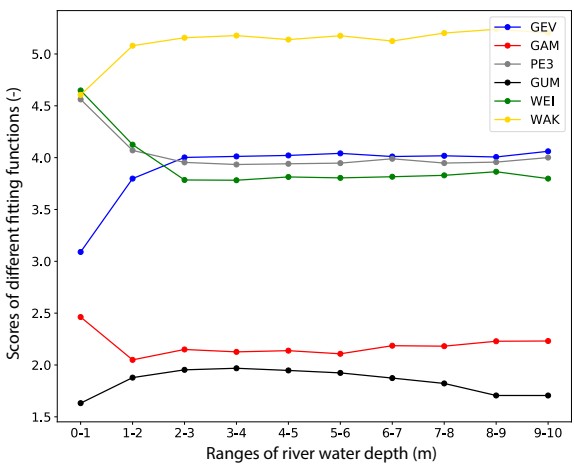

**Figure 5.** The averaged ranking among the different fitting distributions. The results are grouped by grids with different river water depth.

### 3.3 Comparison between different *Runoff inputs*

In the aforementioned analysis, the fitting distributions mainly associate the shape of the distribution of high values. While the runoff inputs mainly determine the average states and have small impact on the fitting performance. Figure 6 shows the ranking of runoff inputs in terms of the mean values of the annual maximum river water depth in the original CaMa-Flood outputs. The spatial variation of the ranks is more complicated than Figure 4, thus the coverage of each runoff input is displayed besides the map in the unit of percentage.

Large variations are found among different runoff inputs in a different ranking. In Figure 6-a, e2o_anu and e2o_univu provide the lowest estimation of the maximum water depth in most regions in the world, except North America, the southeastern Asia and the Green Land (where e2o_ecmwf is the lowest). The regions with the two runoff account for 33.7% and 26.3% of the total continental grids, respectively. In Figure 6-b, e2o_cnrs (48.9%) provides the largest value for most of the land in North America, high latitudes in Europe and Asia, southeast Asia, central Africa and southeastern Australia. Then e2o_univk (24.9%) is the highest in the regions around the Mediterranean Sea and central Australia. e2o_ecmwf (6.2%) provides the highest floodplain water depth in the Amazon River basin. Regarding the runoff inputs ranking in the middle (Figure 6-c), the variety increases as no runoff input is the middle one for a large extent. The coverage of different runoff inputs ranges from 6.2% to 19.7% with a smaller variation than that for the lowest or highest estimates. In regional scale, the e2o_anu and e2o_cnrs are in the middle for most of the river channels in the Northern Hemisphere and south of the 10°S. In the low latitudes in the Northern Hemisphere and the tropical regions, the e2o_univu and e2o_ecmwf are probably being in the middle.

Therefore, for global-scale studies, there is no preference of runoff selection. Ensemble simulation is suggested to account for the different ability of the land surface or hydrological models in different climates or topographic conditions. While for regional studies, observations are recommended to validate the simulations. Ensemble simulations driven by all the runoff

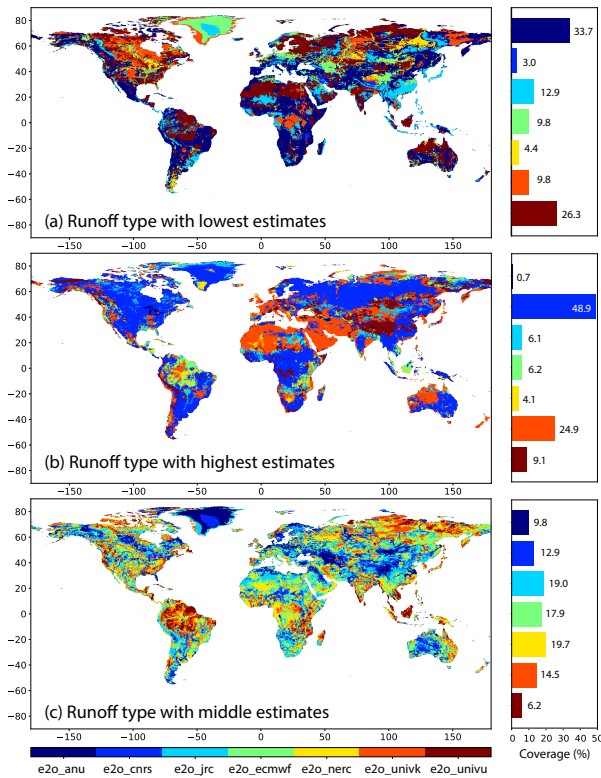

**Figure 6.** The types of runoff inputs corresponding to (a) the lowest, (b) the highest and (c) middle estimates of the mean annual maximum river water depth. The coverage of each runoff corresponding to each map is shown in the right in unit of percentage.

inputs are also preferable if the observation is lack. Otherwise, selecting the runoff inputs which provides the estimation of water level in the middle would be recommended to reduce the risk of large deviation in one single runoff input.

## 4   Regional uncertainties analysis

In this section, the uncertainty range in the water level and inundation due to the selection of investigated variables, fitting distributions and runoff inputs will be discussed. Different from the previous analysis, the results in this section are based on the original results on the FFA, rather than the results after normalization. The uncertainty analysis is concentrated on the lower Mekong region, where the delta is vulnerable to floods. Point analysis and analyses on regional maps are combined to better illustrate the uncertainties from various sources.




## 4.1 Point analysis

A specific point (105.00ºE, 11.54ºN located one grid after the confluence of the main Mekong River and the outflow from Tonle Sap Lake, see the yellow cross point in Figure 8-a) was selected to analyze the uncertainties in the floodplain water depth. The estimated mean water depth as well as the uncertainty range (doubled standard deviation) among different conditions are

shown as the solid line and shaded area, respectively, in Figure 7. The overall mean value of the estimated water depth is shown in Figure 7-a. The water depth at 50% is 6.6m and it is 7.9 m for the 100-year return period flood (hereafter 100-yr flood). The overall uncertainty range is large up to 1.3 m and it is generally the same for different return period frequency (from 0.1% to 99.9%).

In Figure 7-b, the differences between mean floodplain water depth using river depth ($V1\_rivdph$) and storage ($V1\_sto2dph$)

is very small. The uncertainty range is still as large as that in Figure 7-a, indicating that the uncertainty is little contributed by the variables for FFA but other sources. Similarly, subtracting the uncertainty from fitting distributions does not apparently decrease the uncertainty range (Figure 7-c), indicating that the uncertainty resulted from the selection of fitting distribution is still small. Particularly, the mean value for GUM function in the tails of the floods (more than 20-yr flood) is higher thaV2_n results of other functions, indicating that GUM may provide a relatively deviated estimate of mean floodplain water depth for

the extreme flood events. The ranges of other uncertainties in GUM is still similar to the magnitude of uncertainties for other fitting distributions, indicating that the uncertainty from experiments other than the fitting distribution is still large.

Figure 7-d separates the uncertainties of the runoff inputs from the overall uncertainties. It is notable that the mean values significantly vary from different runoff inputs (solid lines in Figure 7-d). For the 100-yr flood, the mean water depth ranges from 6.9 m in e2o_univk to 9.8 m in e2o_cnrs (2.9 m in difference). As for each of the runoff, the uncertainty caused by other

sources (variables and fitting distributions; the shaded area in Figure 7-d) is now very small especially within the normal period (5-yr flood and 5-yr drought) covered by the modelled simulations (35 years in this study). While the uncertainty range starts to increase for the extreme floods. The uncertainty range increases to 0.3-0.5 m for 100-yr flood (on average 25% of the total uncertainty) and 0.8-1.0 m for 200-yr flood (on average 33.3% of the total uncertainty). Though, the uncertainty range is still much smaller than the deviations of the mean values. The increasing uncertainty is similar at the other end of the tails.

The above results demonstrate that runoff input is the primary source of uncertainty to the river water depth simulation. The uncertainty is mainly due to the systemic bias in the runoff inputs. While for a specific runoff input, the uncertainty is small especially during the normal period when the estimated values are available (35 years simulation in our case). In the tails that extrapolation is applied to FFA, the uncertainty range gets increasing mainly due to the different tail shape of various fitting distributions. But the uncertainty range is still smaller than the deviation between results driven by different

runoff inputs. Therefore, for impact assessment over the extreme events, the runoff inputs or the average state of the extremes should be evaluated first with observed information if allowed. Then attention can be given to the selection of different fitting distributions if observations of large floods can be used to optimize the fitting performance especially in the tails.





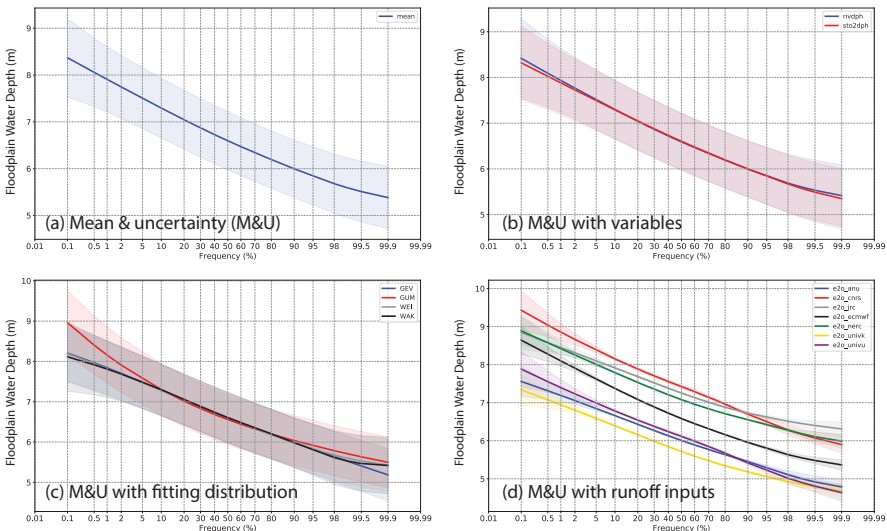

**Figure 7.** Uncertainties in the estimated floodplain water depth at a specific point (105.00°E, 11.54°N) in different groups. a) the mean floodplain water depth and overall uncertainty; b) the mean and uncertainty in groups of different variables for FFA, the uncertainty is then not related to the selected variable; c) the mean and uncertainty in groups of different fitting distributions, d) the mean and uncertainty in groups of different runoff inputs.

## 4.2 Regional analysis – floodplain water depth

The floodplain water depth at a 100-yr flood was first downscaled to high-resolution map (90 m) to show the details with topography and the mean floodplain water depth for the lower Mekong is shown as Figure 8-a. The largest water depth (>10.0 m) is found in the centre of Tonle Sap Lake and the main channel of the Mekong River. Large extent in the lower Mekong delta
5  is suffering relatively low inundation water depth (in dark red). The low water depth is also occurring along the boundaries of lakes and main channels. The river tributaries are also with low average water depth among all the experiments. In summary (Table 2), the inundation area (water depth >0.01 m) of the study area during 100-yr flood is 68809.1 km$^2$. Among which 22.2% of the area is with high water depth (>5.0 m, 15129.5 km$^2$). 33.1% of the inundation area is with water depth less than 1.0 m and 8.8% with water depth under 0.1 m.

**Table 2.** Inundation area with the floodplain water depth corresponding to 100-yr floods (Figure 8-a) with different water depth categories.

| | Floodplain water depth [m] | | | | | |
|---|---|---|---|---|---|---|
| | all | 0.01-0.1 | 0.1-0.3 | 0.3-0.5 | 0.5-1.0 | 1.0-5.0 | >5.0 |
| all [km$^2$] | 68809.1 | 6078.3 | 5030.6 | 3707.9 | 7976.1 | 30886.7 | 15129.5 |
| percentage to all [%] | 100.0 | 8.8 | 7.3 | 5.4 | 11.6 | 44.9 | 22.0 |

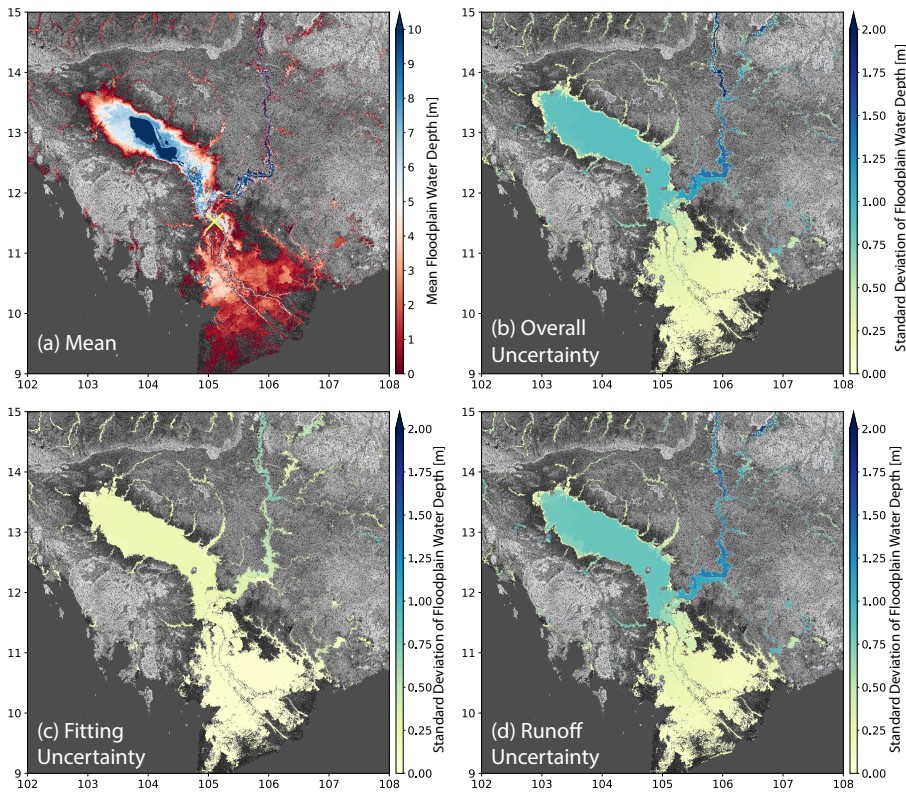

**Figure 8.** (a) Ensemble mean water depth among all the experiments for 100-yr flood and (b) the overall uncertaint (standard deviation) for all experiments. (c) and (d) illustrate the average uncertainty for fitting distributions and runoff inputs, respectively. Area with floodplain water depth less than 0.01 m are masked out.

Figure 8-b shows the uncertainties resulted from different experiments listed in Table 1 except for fitting distributions of GAM and GUM because of their poor fitting performance. In general, the uncertainty range is higher where the estimated water depth is higher (Figure 8-a) as the lowest uncertainties are found in the lower Mekong delta and largest uncertainties in the main channel of Mekong with magnitude higher than 2.0 m. The uncertainty in the Tonle Sap Lake is homogeneous with a
5  magnitude around 1.0 m.

The overall uncertainties mainly result from the fitting distributions (Figure 8-c) and the runoff inputs (Figure 8-d). Whilst the uncertainties from runoff inputs contributed the most because the magnitude in Figure 8-d is very similar to the overall uncertainties (Figure 8-b). This is consistent with the conclusion from point analysis in the previous subsection. It further strengthens the point analysis and makes it valid over the entire region. The uncertainties of fitting distributions are small in
10  the lower deltas, but it gets larger when the water depth increases. The largest uncertainty is approaching 1.0 m in the upper Mekong reaches of the study area. Though, the increases in water depth will not lead to significantly increase inundation area.





### 4.3 Regional analysis – inundation agreement

Despite the uncertainties with estimated mean floodplain water depth, the agreement on the prediction of inundation among different FFA settings might be more important because the inundation will cause damages regardless of the water depth. High inundation agreement will provide the confidence of adapting corresponding actions, for example, evacuation in the most

serious condition. Figure 9 illustrates the inundation agreement among all the experiments in the study area. In general, the agreement is high for the lakes, river channels and the lower Mekong deltas (coloured in dark blue, ∼100%), indicating that if suffering a 100-yr flood, all these regions will be inundated regardless of different runoff inputs and used fitting distributions. Lower agreement is generally with lower estimated water depth around the boundaries of lakes and other inundation areas. Particularly, the large area of the Krong Prey Veng (white square in Figure 9) is with inundation agreement around 50%,

indicating that the resulting inundation is not highly consistent, which means selection of different runoff input and fitting distributions will lead to big differences (inundation or non-inundation).

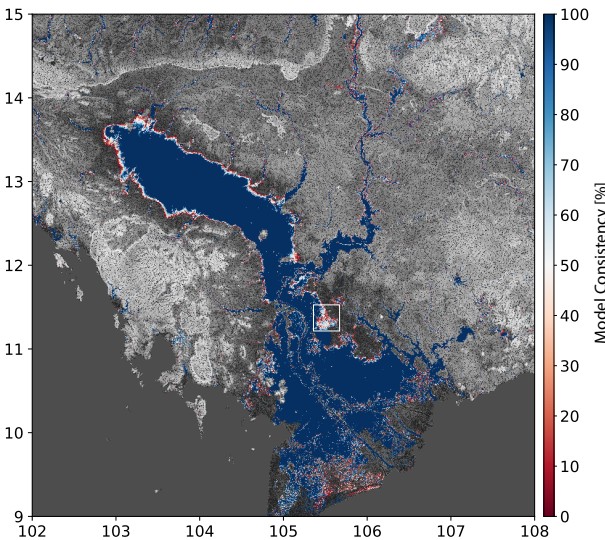

**Figure 9.** Inundation agreement of the estimated inundation (>0.01 m) for the ensemble 100-yr flood. It is calculated as the ratio of the number of experiments which predict the inundation to the total number of experiments. The white square represents the region of Krong Prey Veng.

    Figure 10 shows the inundation area in different categories of mean floodplain water depth. Meanwhile, the inundation area is separated according to the level of inundation agreement, with ≥50% as high agreement and <50% as low agreement. For the regions with high predicted inundated water depth (>1.0 m), different experiments are highly consistent as all the inundation

area is with agreement larger than 50%. In other words, more than 50% of the experiments predict inundation at this location. On the contrary, the agreement is low for inundation area with mean floodplain water depth less than 0.1 m, as 83.8% of the area has a agreement less than 50%. The percentage of area with low inundation agreement decreases from 83.8% for low





mean water depth [0.01 – 0.1 m] to 5.1% for the area with water depth [0.5 – 1.0 m] and to 0% for the area with water depth larger than 1.0 m.

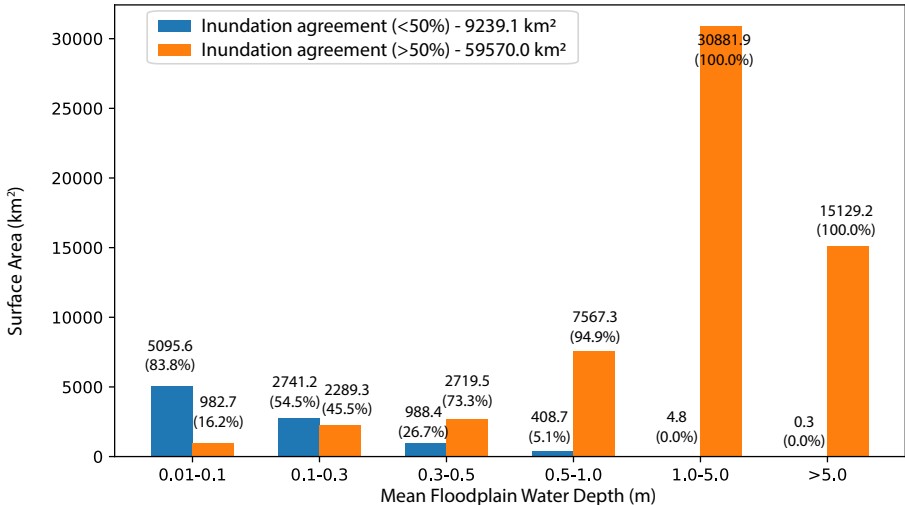

**Figure 10.** Area ($km^2$) and proportions (%) of the inundation with different uncertainty in different categories of the ensemble mean floodplain water depth for the ensemble 100-yr flood. For each water depth category, low agreement and high agreement are divided by criterion of 50%. The labels for each bar is the inundation area ($km^2$, up) and the percentage of the area (%, low) in each category of floodplain water depth.

To conclude, for a potential 100-yr flood, 13.4% of the predicted inundation area (9239.1 $km^2$) is with low model agreement less than 50% as half of the experiments/FFA settings predict non-inundation for this location. Selection of the appropriate

experiments will become more important for flood prediction and risk analysis. For regions with high model agreement (>50%), the adaptions to the predicted floods have to be taken in high confidence. However, the required actions can be different for different flood water depth.

### 4.4   Regional analysis – inundation area

In addition to the 100-yr flood, the predicted inundation area, as well as the uncertainty for all the return periods, are investigated

in this subsection. The mean inundation area averaged over the estimated inundation area in each experiment and the uncertainty of the inundation area are plotted as Figure 11. The mean inundation area increases from a normal flood (return period as 50%, 52135.2 $km^2$) to 62234.8 $km^2$ corresponding to a 100-yr flood (Figure 11). However, it is notable that the inundation area for 100-yr flood is 68809.1 $km^2$ (10.6% higher) in Table 2 if inundation is calculated according to the mean floodplain water depth averaged over different experiments. This difference is mainly caused by the different ways of estimations, as in this

subsection, the inundation area is estimated first for each experiment and then they are averaged to reach the mean value. While for the previous estimates in Table 2, the inundation area is calculated by the averaged floodplain water depth over



multiple experiments. In that case, one single experiment with very high floodplain water depth can lead to high mean water depth (>0.01 m) even if the other experiments do not predict inundation in the same location. This is why, in the Figure 10, the model agreement is very low especially for the area with the ensemble mean floodplain water depth less than 0.1 m.

Similar to the features of floodplain water depth at the selected point (Figure 7), the magnitude of uncertainty range in the
inundation area is similar for all the return periods (Figure 11-a). The uncertainty range for the two ends of tails is a little bit larger. The uncertainties are also mainly resulted from the deviation of means values in different runoff inputs (Figure 11-b). The predicted inundation area for a 100-yr flood ranges from 56000 $\mathrm{km}^2$ to 70000 $\mathrm{km}^2$ in different experiments, indicating a 20% difference to the largest extent.

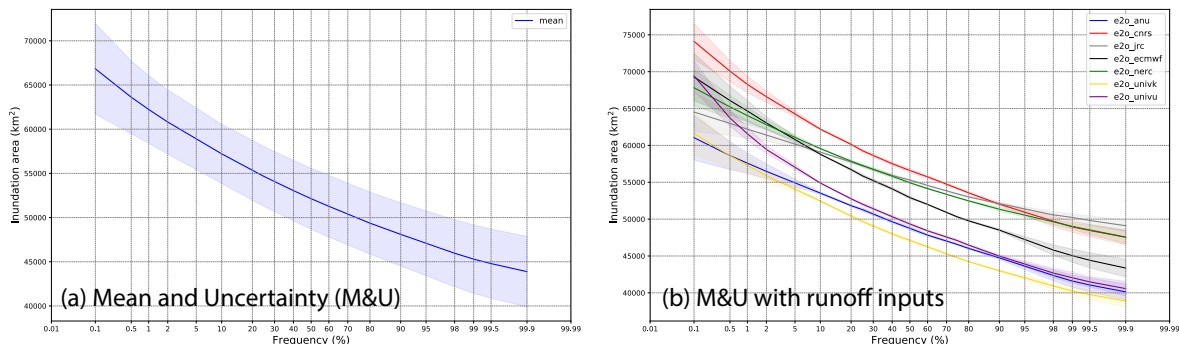

**Figure 11.** The uncertainties in the estimated inundation area for the study area. a) shows the mean inundation area and the overall uncertainty, b) shows the mean and uncertainty related to runoff inputs.

### 4.5    Validation with other results

In this subsection, the inundation map is compared to two flood hazard maps from different sources. The first source of the flood hazard map is from GAR (Global Assessment Report on Disaster Risk Reduction, GAR, 2015). This dataset was observation-based for large rivers. Quantiles of the river discharge were estimated based on the collected stream-flow or proxy data from homogeneous regions. The calculated quantiles were then introduced to river sections with topographic data (SRTM) and a simplified approach based on Manning's equation to model the water levels (Rudari et al., 2015). The second source is JRC
(Joint Research Centre Data Catalogue) data based on streamflow data from the European and Global Flood Awareness System (EFAS and GloFAS) and computed using two-dimensional hydrodynamic models-CA2D (Alfieri et al., 2014; Dottori et al., 2016). Though, it was already mentioned in the references that there are limitations in the model and the maps might differ from official flood hazard maps. The two maps are plotted as Figure 12-a and 12-b.

The spatial resolution of the GAR dataset and JRC dataset is 30 arcsec. It is recommended that the comparisons are conducted
on the same spatial resolution. We therefore downscaled the original CaMa-Flood results to 30 arcsec (Figure 12-c). Large differences of the inundation area in the tributaries can be seen by comparing the three maps in the lower Mekong River,

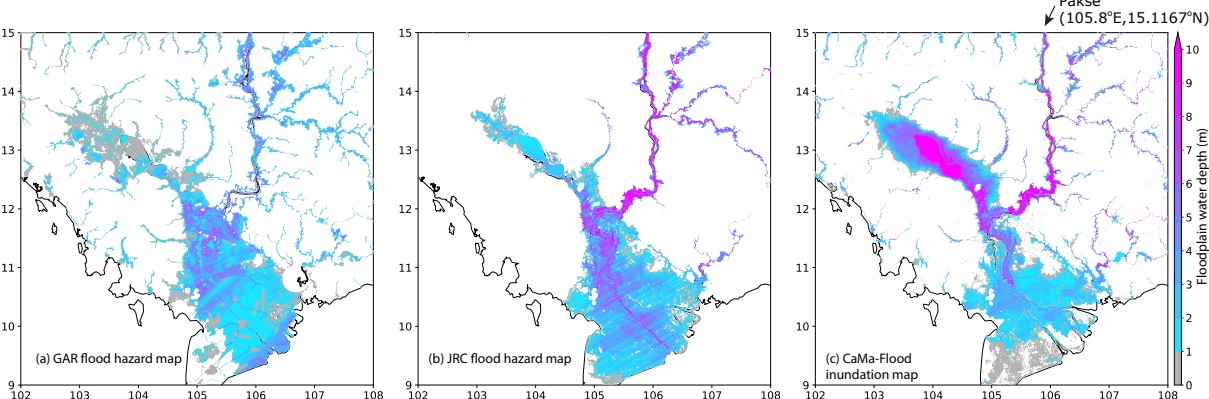

**Figure 12.** Comparison of inundation map with other flood hazard map for 100-yr flood (a) GAR, (b) JRC, (c) CaMa-Flood. The three maps are all at 30 arcsec spatial resolution. Pakse is one of the GRDC gauges just upstream of the study area of the lower Mekong River.

as almost all the tributaries are inundated in GAR dataset, while very few tributaries are in high risk in JRC dataset. The risk of inundation in tributaries is in the middle for CaMa-Flood result. Flood extent simulation in tributaries is affected by many factors including model's spatial resolutions, model parameters such as channel cross-section, and also flood frequency analysis in observation based product. The other difference is the floodplain water depth: both the two model-based results (JRC and CaMa-Flood) predict similar but much higher water depth (>10 m) along the main river channels. While the water depth in the GAR data is only around 5 m. The water depth in the Tonle Sap lake is probably not considered as flood hazard in GAR and JRC. The third difference is related to the topography. Stripes are found in the lower Mekong delta regions in JRC, which is caused by the biases of the topography (this has been well explained and addressed by the MERIT dataset, see reference of (Yamazaki et al., 2017)). In GAR, the spatial distribution of the water depth over the delta regions are also not vary realistic. The accuracy of the topography (DEM) is of vital importance for inundation calculation especially for the lower flat regions. The flood extent with water depth between 0.01m-1.0m is the most sensitive to the topography (grey area in Figure 12). Corresponding extent in GAR is mainly over the Tonle Sap lake, where the underwater bathymetry is not accessible and the backwater effect is difficult to accurately modelled; the corresponding extent in the CaMa-Flood is mainly distributed in the coastal deltas because of the bifurcation channels (Yamazaki et al., 2014) and around the boundaries of the flood extent which seems more realistic. Despite of the differences, CaMa-Flood hazard map (Figure 12-c) is reasonable since its extent is almost within range of existing hazard maps (Figure 12-a,b) even though the baseline topography or methodology are different. The floodplain water depth in CaMa-Flood also exhibits similar spatial patterns especially with the JRC flood hazard map, despite of the values in the Tonle Sap lake.

A summary of the total inundation area based on the three maps is provided as Figure 13-a, with two different threshold for inundation as 0.01 m (left panel) and 1.0 m (right panel), respectively. For either of the criterion, the sum of the inundation area in CaMa-Flood (averaged of different experiments) is within the value range of GAR and JRC. The flood extent with 0.01-1.0

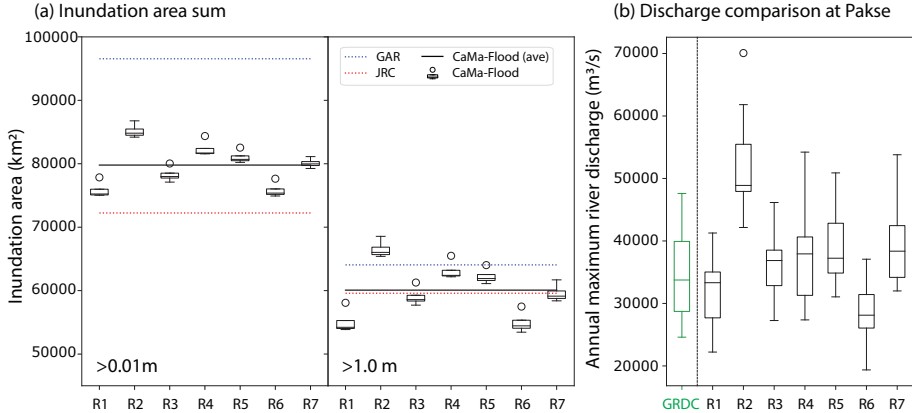

**Figure 13.** (a) shows the uncertainties of CaMa-Flood total inundation area driven by different runoff (boxplots, from R1 to R7, Table 1) and the comparisons of average area with other two maps (blue dashed and red dashed line). The sum of the inundation area is counted by area with floodplain water depth higher than 0.01 m (left panel) and 1.0 m (right panel), respectively. (b) Comparison of the estimated annual maximum discharge at Pakse gauge by CaMa-Flood driven by different runoff inputs. Pakse gauge (105.8°E,15.1167°N) is located right up to the edge of the upper reaches for the study area (see Figure 12-c). The plot is only for the annual maximum daily discharge covering 1980-1993 determined by the available period of GRDC.

m water depth accounts for around 25% for the three products and contributes the most to the overall deviation. While for larger flood with flood water depth larger than 1.0 m, the three products provide similar total inundation area. The remaining differences can be explained by the runoff uncertainty because the deviation of results driven by different runoff inputs exceeds the difference among the mean value of the three products (right panel of Figure 13-a).

5    The variation of the inundation area is closely related to the variation of the predicted discharge at Pakse which represents most the river discharge to the lower Mekong from upstream (Figure 12-c). The discharge driven by R2 (e2o_cnrs) is the highest and R6 (e2o_univk) is the lowest in both the inundation area and the discharge (Figure 13-b), while the real discharge (GRDC) is among the spread ranges of all the seven different runoff inputs. It is also notable that the inundation area for CaMa-Flood at 3 arcsec is 68809.1 $km^2$ and 46016.2 $km^2$ (Table 2) for area with water depth larger than 0.01 m and 1.0

10   m, respectively. The total inundation area is around ~20% lower than the results based on 30 arcsec resolution shown in (Figure 13-a, 79791.9 $km^2$ and 60073.7$km^2$, respectively). The difference at the two spatial resolution is resulted from the ability to describe the heterogeneity in topography with the high-resolution topography data. This large difference indicates that the current assessment on the flood risk/impact could have been overestimated and it requires a necessity to apply a much higher-resolution topography to the current methodologies.



## 5 Overview of uncertainties for different floods at a global scale

Figure 14 summaries the mean floodplain water depth and the related uncertainties over the globe in order to have a broader view of FFA above regional results. Results corresponding to floods in two different return period (i.e., 100 year and 20 year) were compared. 20-yr flood is within the computation period (35 years in this study) while the water depth for 100-yr floods

has to be obtained through extrapolation.

For the mean values, the floodplain water depth will exceed 10 m in most of the main channels of large rivers if suffering a 100-yr flood (Figure 14-a). The risk is especially high in the Amazon River, Congo River, large rivers in southern China, southeastern Asia and the rivers in Siberia. The spatial patterns of the floodplain water depth for 20-yr flood are very similar to the 100-yr flood but with lower magnitudes (Figure 14-b). The total uncertainties, including different fitting distributions

and multiple runoff inputs are shown as Figure 14-c,d for 100-yr and 20-yr flood, respectively. In general, high uncertainties are along with high predicted water depths. What needs to be noticed is the higher uncertainties tend to occur in mountainous regions rather than the flat regions, typically in the rivers originating from the Tibetan Plateau and Siberia. The uncertainties in the lower Yangtze, Mekong, Salween rivers are much lower than estimated uncertainties in their upstreams while the mean water depths are higher in the lower reaches. High uncertainties are also found in the Congo River. Although it is not surrounded

by high mountains, its main channel is relatively short compared to its drainage area. The fluctuation in the river discharge will lead to high gradients in the water level which could be associated with its high uncertainties shown in (Figure 14-c and d). The uncertainties in the Amazon is relatively low compared to the high mean water depths, this could result from the higher consistency of modelled runoff or fitting distributions and its relatively flat topography. The uncertainties in 20-yr show relatively lower values than that in the 100-yr flood.

As stated in the regional studies, the uncertainties of predicted water depth are mainly contributed by the deviations in the runoff inputs. At a global scale, the same results are found that the uncertainties resulted from fitting distributions contribute much less than that from the runoff. Figure 14-e and f show the uncertainties due to selections of the fitting distributions in the experiments of group driven by e2o_ecmwf. Very small values are found in the same scale of colours for the total uncertainties. On the contrary, the uncertainties due to runoff inputs (within experiments group fitted by GEV distribution as an example,

Figure 14-g,h) have similar magnitude of the total uncertainties, indicating that most of the uncertainties can be attributed to the deviations in runoff inputs.

The other finding is for the flood within the period of simulations (20-yr flood in the case of 35-years' simulation in this study), the uncertainties due to fitting distribution is nearly zero for the globe (Figure 14-f). While, higher uncertainties can be visualized for the 100-yr flood (Figure 14-e) especially in the Amazon, Indus and rivers in the southeast Asia, southern China

and the Siberia. The differences tell that the uncertainties due to fitting distribution are mainly because of the extrapolation out of the period with simulated or observed data. While within the period of available data, uncertainties due to fitting distributions will be efficiently constrained.

Figure 15 shows the changes in contribution of uncertainties in fitting distribution (and runoff inputs) to the total uncertainties from 20-yr flood to 100-yr flood. Positive values show that the uncertainty contribution is higher in 100-yr flood than that in 20-


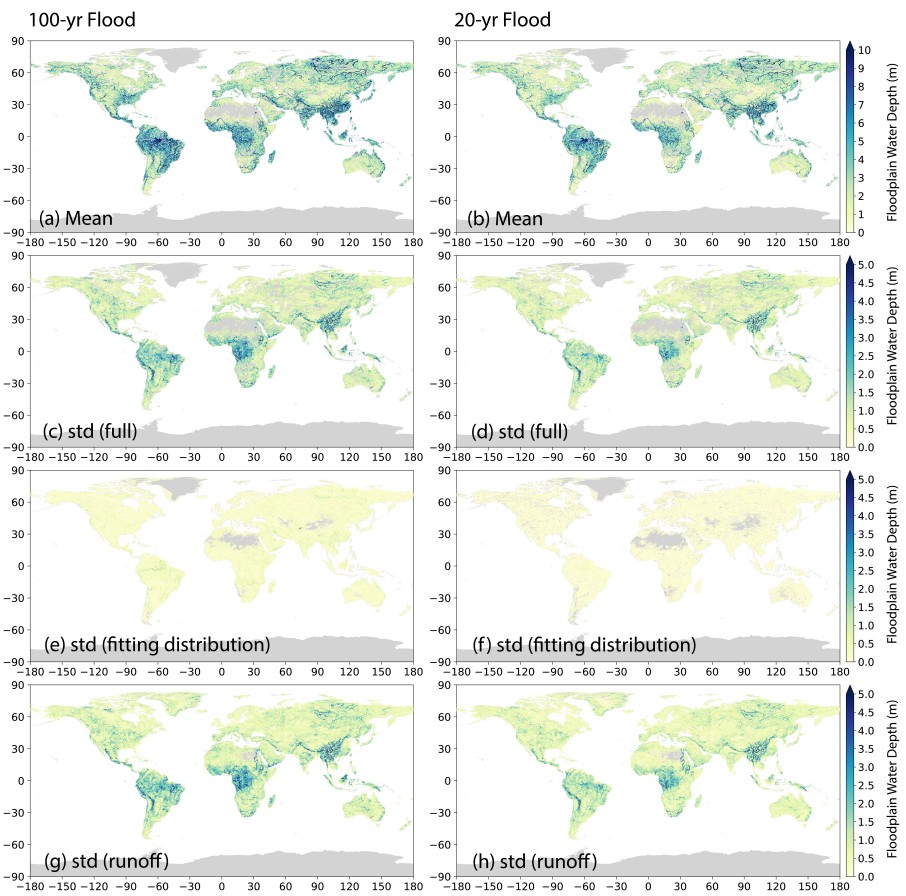

**Figure 14.** The mean floodplain water depth and uncertainties over the globe for 100-yr flood (left column) and 20-yr flood (right column). The first row is the mean floodplain water depth. The other three rows are the total uncertainties and uncertainties from fitting distribution (with e2o_ecmwf runoff) and runoff inputs (with GEV fitting function).

yr flood. Figure 15-a indicates that for almost all the global grids, the contribution of the uncertainties due to fitting distribution increases. This is mainly due to the extrapolation of fitting and larger uncertainties occur in the fitting tails. Although the uncertainties in runoff inputs are still the dominant, we need to pay more attention to the selection of fitting distribution for the rarer floods.

5    Figure 15-b indicates the changes in contribution of uncertainties from runoff inputs. Obvious deviations are found between the mountainous/dry regions (e.g., The Rocky Mountains, Sahel, Central Australia, Central Asia) and floodplain/wet regions (e.g., Amazon, Congo, Ganges, Indonesia). The change means, for the rare floods (100-yr flood), the contribution of uncertainties from runoff inputs increases for the wet regions. Selection of the appropriate runoff becomes more important as well in this case. Note that the total uncertainty is not equal to the sum of the uncertainties from runoff and fitting distributions, the

10    increases of both contributions indicate higher necessity of accounting for the uncertainties especially in the wet regions.

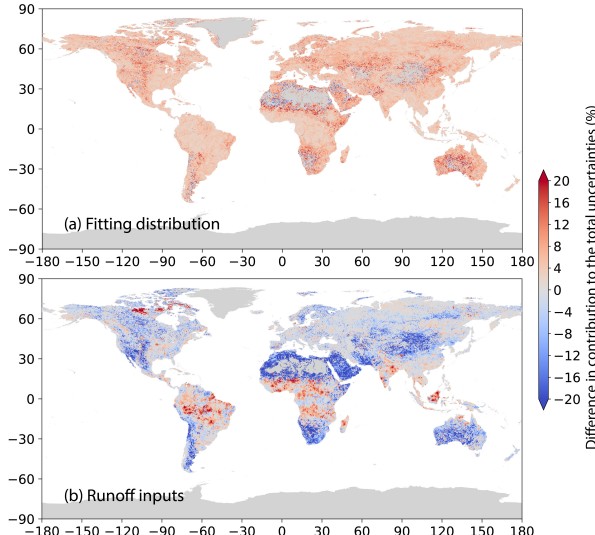

**Figure 15.** Changes in the contribution of uncertainties to the full uncertainties for 20-yr flood and 100-yr flood. a) represents the contribution of uncertainties among fitting distributions and b) represents the contribution of uncertainties among runoff inputs. The contribution is calculated as the ratio of standard deviation to the total uncertainties of multiple experiments, corresponding to the third (and fourth) row and the second row, respectively.

## 6   Discussion and Conclusion

### 6.1   Discussion

This study assesses the flood risk based on pure simulations with a global hydrodynamic model (CaMa-Flood). Due to the multiple choices of runoff inputs, fitting distributions for flood frequency analysis (FFA) and the variables for FFA, the analysis

of flood risk can be uncertain. The performance of the FFA is also varying in experiments with various combination of the above conditions. We conclude from the analysis with a performance metric $aic$ that the river water depth (V1_$rivdph$) is more suitable for the FFA than converting the river storage to river depth (V2_$sto2dph$). Applying F2-GAM and F4-GUM is not suitable for the FFA because it leads to low fitting performance at a global scale through a ranking approach. F6-WAK performs the best mainly because it has five parameters while other 3-parameters fitting distributions (F1-GEV, F3-PE3 and

F5-WEI) are in the middle. The fundamental limitation of this study is that the conclusion is based on uncalibrated simulations rather than observations and conclusions can differ if different routing method (rather than CaMa-Flood) is used. However, the attempts to assess the flood risk at a large scale provide reference information and routines for similar analysis. Moreover, FFA at the global scale based on observations is not feasible because of the lack of long-term observations on the water level or water storage and the difficulty to reach all the existing data. Model simulation has its advantage as it covers a larger area and longer

period because the forcing variables are much better in temporal and spatial coverage (Jones and Kay, 2007). Simulations can





also estimate the high floods while the *in-situ* measurement for the large discharge can be with high uncertainty as well (Di Baldassarre and Montanari, 2009). Thus at the current stage model estimations are still important and can be relied on as a good reference data to conduct the flood frequency analysis if the uncertainties are properly treated.

The uncertainties in the estimated floodplain water level can be derived from multiple sources. Trigg et al. (2016); Bernhofen et al. (2018) compared the flood hazard maps (inundation area) over Africa with multiple products from six different institute. Large diversity is found in the inundation area among different model products, however, they have difficulties to attribute the variations or explain the behind reasons because different products use different forcing input, different model, different topography and different frequency analysis. They suggested rather than the product-level comparisons, component-level comparisons with limited variables could be better to attribute the uncertainties. In this study, by fixing the hydrodynamic model (CaMa-Flood), other uncertainties can be more easily quantified and attributed. Runoff inputs are regarded as the largest contributor to the final uncertainty. Because the runoff inputs are driven by the same WFDEI forcing, the differences in the output therefore explicit the difference of land surface models or hydrological models (Schellekens et al., 2016). Moreover, FFA only uses the maximum water level (or water storage), the variety in the FFA only demonstrate the performance rainfall-runoff models in reproducing the discharge peaks. Separation of surface runoff and subsurface runoff and the evaporation rate during the extreme raining events can lead to the differences in total runoff and the hydrodynamic processes during routing. Among the different runoff inputs, the e2o_ecmwf by HTESSEL stands in the middle according to the point analysis in Figure 7-d. However, the runoff input providing the middle estimation of water depth varies for the world (Figure 6). This shows no runoff input is preferable at a global scale for estimating the high water states, nor the land surface or hydrological models. Either the estimation needs validation with observations (Lin et al., 2019), or ensemble simulation is needed at a large scale (Warszawski et al., 2014; Schellekens et al., 2016). For specific regions, Figure 6-c can be used as a reference for selecting a runoff input in the middle to reduce the risk of large uncertainty in floods if conducting all the runoff inputs consumes a large amount of computation.

The agreement on the inundation prediction is assessed for the lower Mekong basin. In general, the agreement is high (>50%) for most of the inundation area (86.6%), and approaching 100% if the predicted mean floodplain water depth is larger than 0.5 m. Despite the highly certain inundation area, attention should be also paid to regions where the inundation is predicted while the model agreement is relatively low (<50%) because the uncertainty from multiple sources will cause different consequences (inundation or non-inundation), especially the low human habits as shown in Figure 9. However, both the regions with different inundation agreement are in high risk if suffering large floods, different reactions should be taken in different priority according to the water depth and agreement on inundation as well as the local conditions regarding population and property. Inundation area is also calculated. We would like to note that, because of the different ways of estimating the inundation area, there is a 10% difference in the total inundation area for a 100-yr flood. The difference is mainly due to the low model agreement over the regions with the low predicted floodplain water level. The variations of the inundation area by different experiments can be as large as 20%. The ratio can be higher if excluding permanent water bodies such as the lakes and river channels. This necessitates large effort to decreases the uncertainties related to the FFA. Assimilation of the altimetry data (Revel et al., 2018)


or inundation area (Hostache et al., 2018) from satellites can be a solution to evaluate model results at a large scale especially in regions where ground observations are not available.

Two other sources of flood hazard maps are utilized to validate the results from CaMa-Flood. However, because the two maps are also generated by models to some degree, we cannot quantitatively evaluate the CaMa-Flood performance from the comparison. The differences are mainly in inundation area with shallow water depth and in the tributaries. While for larger floods (water depth >1.0 m), the differences among three products are much smaller and the variation of CaMa-Flood results from the variations of runoff inputs. The variation is related to the discharge estimation at the upper gauge. However, the proper validation of the inundation at a theoretical level (100-yr flood) is still not feasible because of the lack of spatial observations for the equivalent flood. Comparison with other products only increases the credibility of estimations but helps to identify the discrepancies among different products. To some degree, CaMa-Flood is superior in describing the spatial patterns and is more flexible in the selection of different spatial resolution. The variation between results driven by different runoff inputs requires an improvement of runoff estimation through bias correction or data assimilation.

During the validation, we found a ~20% deviation of the inundation area between two different spatial resolution (3 arcsec and 30 arcsec). Similar result is found in Hinkel et al. (2014) as they evaluated the coastal flood damages by using two kinds of topography data GLOBE (30 arcsec) and SRTM (1 arcsec). The exposed area, population and assets were lower by 50% to 70% in assessment with high-resolution topography (SRTM) than the low-resolution GLOBE. Therefore studies with 30 arcsec (1km) (Jongman et al., 2012; Ward et al., 2013; Jongman et al., 2015) could provide overestimated results for this end. This difference requires us the ability to have higher resolution topography and corresponding technologies to obtain the results with more spatial details.

The floodplain water depth and its uncertainties are investigated at a global scale. The conclusions are similar to regional studies as that the major contribution to the final uncertainties resulted from the deviations of runoff inputs. Although the uncertainties in the inundation area is yet investigated, the results will be in the same direction since the inundation area is highly associated with the floodplain water depth. Comparisons of the contribution of uncertainties from fitting distribution and runoff inputs to different floods (e.g., 100-yr, 20-yr) indicates that uncertainties from fitting distributions resulted from the extrapolation out of the period with data. Having longer-term modelled or observed data will greatly reduce the uncertainties. Investigation of historical floods will also benefit for the improvement of FFA (Payrastre et al., 2011). The deviating results about the uncertainties contribution from runoff inputs for wet and dry (or mountainous and flat regions) also requires a differentiating treatment to different kinds of floods in different regions. The behind reasons for the differences are associated with the topography or model performance in different regions, while they remain to be investigated.

The water depth, flood risk and damages are sensitive to the flood protection adaptions (Ward et al., 2013). Dam regulation (Wang et al., 2017) and river levees (Berning et al., 2001) are effective ways to mitigate the potential risks of floods. However, the database is not accessible at a large scale and the flood protection is not applied in the current model. Attempts on the improvement of CaMa-Flood by integrating the dam regulation (Shin et al., 2020) and levees (Tanaka and Yamazaki, 2019) have been tested at a regional scale.





## 6.2 Conclusions

This study assessed the uncertainties in floodplain water depth after flood frequency analysis. Uncertainties can result from the selection of variables for FFA, fitting distributions and the runoff inputs which drive the routing model for estimating the water depth. Among all uncertainty sources. Uncertainties from the runoff inputs contribute the most to the total uncertainty, mainly

due to the deviated mean values of extreme water depth. This suggest the importance of rainfall-runoff model calibration (or runoff bias correction) if gauge discharge observation is available. No preferable runoff inputs are available at the global scale, but the fitting performance implies that directly using the river water depth for FFA is better than using converted water depth from water storage. The fitting distribution WAK is the best among the various fitting distributions. The results of model agreement for inundation estimation is expectable as high agreement is found for inundation regions with high predicted

floodplain water depth. But additional information of model agreement will be helpful for the decision-makers during the flood protection. Inundation area related to the water level also shows large uncertainties, which will increase the difficulty of assessing flood risk and flood damages. The variation of contribution of uncertainties from fitting distribution and runoff inputs for two different level of floods (100-yr flood and 20-yr flood) shows that uncertainty from fitting distribution is due to the extrapolation out of the period with data, and increases with the flood magnitude. While uncertainties from runoff are spatial

varied and the contribution from runoff can be higher for larger floods in wet regions. Overall, model calibration/validation with advanced tools (assimilation of remote sensing products) as well as the model improvement by taking into account the human interventions are needed to reduce the various uncertainties.

*Data availability.*

   The global hydrodynamic model CaMa-Flood is available from http://hydro.iis.u-tokyo.ac.jp/~yamadai/cama-flood/index.
html. The topography data MERIT is available from http://hydro.iis.u-tokyo.ac.jp/~yamadai/MERIT_DEM/index.html. The JRC flood hazard map is available from https://data.jrc.ec.europa.eu/collection/id-0054 and the GAR flood hazard map is available from https://www.preventionweb.net/english/hyogo/gar/2015/en/home/index.html. The estimated floodplain water depth and related source codes are available from the authors upon request. The library *lmoments3* for L-moments parameters estimation is available from https://github.com/OpenHydrology/lmoments3.

*Competing interests.*

   The authors declare no competing interests.





*Acknowledgements.* This study was supported by "KAKENHI 20H02251" by MEXT Japan and also by "LaRC-Flood project" by MS&AD Holdings. The computation is run at the server in the Yamazaki Lab in IIS, The University of Tokyo.



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
