# Peer review of "Uncertainty in flood frequency analysis of hydrodynamic model simulations"

_Natural Hazards and Earth System Sciences, 2020_

## Referee Comment (RC1) · Anonymous Referee #1 · 3 Sep 2020

General comments:

Zhou et al. present a novel and interesting analysis of uncertainties inherent to global hydrodynamic models. Many studies over the past ∼5 years have posited that global flood models produce widely divergent results, yet each of them fails to explain or evidence the cause of this divergence. The authors begin to address some of these previous gaps in the literature and have produced some (though limited) conclusions of genuine interest. However, there are fundamental problems with the methods, analysis, and wider interpretation/discussion of their conclusions that require addressing before publication is to be considered in NHESS. Do not be put off by the volume of comments. I do think the work has great value and is something the field desperately requires. To be the truly impactful contribution this work needs to be, I offer the following (I hope,

constructive) comments. These broadly relate to:

- Model sensitivity vs. uncertainty analysis

- A single model with often extremely limited geographic scope

- Considered uncertainties in the context of 'unconsidered' uncertainties

- The relevance of hydrologic variable choice

- The contextual relevance of distribution goodness-of-fit

- The need for some benchmark data
* * *
Specific comments:

The introduction is mostly good, but could contain a richer discussion on why unstitching the uncertainties of global flood models is needed and how past studies essentially have failed to do this. There are also many sweeping or incorrect generalisations, that may simply be a function of imprecise English (for which I am sympathetic and understanding). This is the case throughout the manuscript (e.g. the sentence in line 26-27 p. 2 makes little sense as the same could be said for RFFA; line 3 p. 23 says the studies assess flood risk, when it does not).

The overarching problem in the introduction is that it makes the reader think some quantification of the uncertainties – validation, against observations – is carried out by the authors, and this is not the case. Truly, this analysis is a sensitivity analysis of 1 model. While I appreciate this illustrates the 'uncertainty one should have about conclusions drawn using this model', really it just tells us what the model is sensitive to and by how much. The paper as a whole needs more of a framing as a sensitivity analysis rather than formal uncertainty quantification.

The methodology, if framed as a sensitivity analysis of CaMa-Flood, appears thorough

and fit-for-purpose. In general, justification of the use of a single model and subsequent analysis in specific regions (even specific grid cells) is needed. How universal are the conclusions in light of these methodological choices? Of course, these are only uncertainties related to the subjective choice of model tests. It is worth stressing that the reported uncertainties make the (of course, incorrect) assumption that terrain, channel bathymetry, human influence, and model parameterisation are certain.

It is not clear why river depth and water storage are chosen as variables of interest – this needs further explanation, as I can not yet see the significance of doing so. Common application of FFA is to discharge, yet this is not done here. Further discussion of the AIC is needed: what constitutes a 'good' result in this context is not specified. Equally, what is the relevance of this metric in terms of model uncertainty? What is the relevance of a good fitting distribution in the context of the uncertainty in the absolute values themselves? Are the authors saying that a variable with a poor AIC contains no relevant information for FFA? Really, it just shows a suitable distribution has not yet been found. I think section 3.1 fails to recognise the variable of choice is arbitrary and depends on the model used and the question asked. We all know that a 100-year rainfall $\neq$ 100-year streamflow $\neq$ 100-year economic loss. So frame this strand of analysis in the context of why the variable you choose matters and why this is interesting.

I can not see evidence that WAK is the best distribution because of it having 5 parameters. As the authors mentioned, it may just be overfit to the simulation record. The reality is we have no idea which distribution we should extrapolate with – and this is not something the AIC can test.

The section 3.3 analysis of runoff is interesting, but the results are stated in such a way that the authors expected the analysis to produce a 'preferable' runoff product. No feature of the analysis performed could identify such a thing. It is not clear why being a runoff product in 'the middle' is the best place to be: it could be that the lowest estimate types are actually best! It is a problem throughout the paper, where a suitable performance benchmark has not been found. Ensure the results are framed and reported as

sensitivities, not as good/bad.

I like the analysis in section 4.1, but I'm not sure why this could not be done for every global grid cell – with normalised results – and presented in the same way. How representative is this grid cell? It may also be interesting here to compare the AIC results to Figure 7c: exploring some of my above comments on why AIC matters more quantitatively (i.e., does high/low AIC [thus, how good the distribution fit is] matter in the context of inundation?)

As for the rest of section 4, the analysis is good. While I appreciate visualising the globe at this scale is difficult, a lot of the calculations could still easily be done globally. It leaves the reader wondering whether different climatologies and geomorphic settings might have different conclusions. Deltas are difficult to model – particularly for models with poor/no representation of coastal boundaries – and so may have distinct features of uncertainty to other areas. I see no reason for the authors not to report findings elsewhere.

I do not see any value to section 4.5. I have little doubt the CaMa-Flood 100-year map is more accurate than the GAR and JRC maps: it is an uninformative comparison, and certainly not "validation". You only have to look at the stripes of JRC's map in Figure 12b to know that that is not a model you should aspire to resemble! I appreciate finding suitable validation data is difficult, but it is difficult to understand the relevance of the authors' conclusions without some. Perhaps running this analysis in the US or western Europe where high-quality models exist and comparing to those would be a good idea.

Section 5 is strong, but will benefit from drawing on some of the above points. Generally, the manuscript is quite long, and so the impact from section 5 is dampened by unnecessarily long analysis in 4.2-4.4. Throughout the paper, I would ensure each test is a worthwhile inclusion for the conclusions drawn. At present, there are many analyses which offer little additional information which I would consider cutting.

Figures are generally good quality, but most need to be larger. I would change the

colour scheme of some figures (e.g. 4-6) where colour scales are used for variables which are not ordinal (no reason to go from blue to red, when the distributions are in no order).
* * *

---

## Referee Comment (RC2) · Anonymous Referee #2 · 14 Sep 2020

General comments

This paper is based on large scale hydrologic-hydrodynamic simulations to investigate different sources of uncertainty in flood risk estimation, with the use of flood frequency analysis tools. The chosen topic deserves some interest, though the analysis is based on a specific configuration of a set of available hydrological model output (from the Earth2Observe project) and an in-house hydrodynamic model (CaMa). However, the focus on the global domain makes it of larger interest for a wider community.

- Among the main limitations of the manuscript is the sub-optimal use of the english language, including both terminology, grammar, typos and structure of the sentences, which makes it hard to read and at times hampers the understanding of the content. I strongly suggest to work and improve it with the help of a native speaker.

[Figure]

- Another important comment is related to the general framing of the analysis. In the current version a number of analyses are performed, focusing on different aspects, though in my opinion it lacks a consistent storyline and some reasoning behind why they were made and clear statements about what we learn following their results.

- The manuscript is too long compared to the information content it brings. I suggest shortening following the comments below. A number of figures should be removed, improved or put in the supplement material, for the reasons I explain below in the specific comments. In particular, I'm speaking about Figures 4 and 5 wrt the issues with fitting analytical functions with different degrees of freedom (comment #10), Fig. 6 (comment #18), Fig. 10, 12, and 14 (comments #24, #27, #29)

Specific comments

1. p2, l8-9: acronyms should be defined with "full name (acronym)", e.g., Global Runoff Data Centre (GRDC). Same for p3, l5 and l26. 2. p2, l14: Pearson type III 3. p3, l1: suggested "connected" –> "analyzed the relation between ..." 4. p3, l3-5: Sentence not clear. Please rephrase. 5. P4, l3: please define the acronym SAR 6. p4, l10: "various runoff inputs" is too general. Please add details here or a reference to the details included in Sect. 2.2 wrt the inputs used. 7. P4, l13: I suggest adding an introductory sentence here to give more details about the experiment itself, before jumping to the uncertainties to investigate. 8. P4, l14-16: please improve this part. Also, I find the variable names V1_(rivdph) and V2_(sto2dph) not very intuitive. Why not simply calling them depth and storage? Especially sto2dph creates confusion on whether it is a storage or a depth. 9. Table 1: I suggest removing "Various" in the caption. 10. P5, l12: Note that the Gumbel and the Gamma distributions have 2 parameters. In fact, results in Figure 5 seems to me the natural consequence of fitting a series of points with mathematical functions with different degrees of freedom, where the 5 parameter distribution is able to fit the data more skillfully (though it doesn't mean it will be more skillful in predictive mode for future floods), Then the 3 parameter distributions and the 2-parameter Gamma and Gumbel as the least skillful. One would obtain similar results

when fitting the series of data with polynomials of grade 5,3 and 2, because higher grade polynomials can fit better the input data. 11. P5, l13: I suggest renaming this section (e.g., "Fitting performance" or similar) 12. p5, l15: calculated 13. p5, l19-20: This should be expressed more clearly. E.g:" Smaller aic denote higher fitting performance" or similar, which is actually better written in p6, l23-24 14. p6, l24-26: Use active rather than passive form (e.g., "we compare") 15. p7, l6-7: Is the normalization the real reason? Also, I suggest giving more details on how to weigh the aic values. What is the optimum? What are normally considered good or bad values? It is not intuitive for those who have never used it. 16. P8, l8: "The later peak" – > "the latter" 17. Figure 3: Interesting to see how the pdfs of gamma and gumbel have similar peaks to the other distributions only for the storage, but not for the river depth. Indeed it is clearly fisible also in Fig. 3c. Would be interesting to investigate and motivate the reasons. Now it is only mentioned but no justification is given. 18. Figure 6: How does this analysis relate to the FFA and to the rest of the paper in general? I'm not sure of the value of these maps, given the little information the readers have on the 7 runoff inputs, and also because there is no clear patter identified. Perhaps the main information one can obtain is that anu and univu tends to be on the lower side, while cnrs and univk on the higher side. Yet, this doesn't say anything about the skills of these estimates, which would imply validation with gauge data at a number of stations. 19. P13, l2: after – > downstream 20. Note that Figure 8 is referenced before Figure 7 21. Sect 4.1 refers to return periods in Fig.7, hence in Fig.7 I advise to show return periods in place of frequencies. In any case, to be correct you should refer to those as annual frequencies of occurrence, to avoid confusion. Also, in Figure 7c, why not all distributions are shown? 22. P14, l2: please give some details and possibly a reference on the downscaling procedure. 23. P14, l3-6: To aid the assessment of water depths I suggest showing in Fig.8 a map or contour of the permanent water bodies. Clearly it is normal to have higher water depths in rivers and lakes, compared to areas normally dry. Also, I cannot find information about the terrain model, in particular whether it represents the river bed or some reference water level. This is important for this analysis.

24. Figure 10: results shown in this figure are rather obvious. I suggest removing this figure as it brings little information. Over large inundation depths it is normal to have good agreement on whether there's inundation or not, as having poor agreement would mean huge differences in the results of the model used (hence very poor skills for some models). 25. P17, l11: return periods should not be expressed as percentage 26. p18, l10 and Figure 12: Is this the mean inundation of the 7 models? Clarify 27. I find the analysis in Figure 12 of limited use, being a qualitative visual comparison with two other publicly available maps, but also resulting from modeling exercise with limited calibration. Similarly, the comments in p19, l14-18 are partly speculative. More rigorous validation with observed flooded areas would give much more strength to the paper. 28. P 21,l6: for flood impact assessment it is more interesting to know (even smaller) inundation depths in areas where people live or where economic assets are, rather than the inundation in the main channels, which has fewer fields of application. 29. Figure 14 is unreadable and of limited use in the present form. It is impossible to get enough spatial details of a global inundation map at such small scales. Furthermore, the left and right column are almost indistinguishable. I suggest removing this figure and rather put it in the supplement, together with a number of inset panels zooming into some areas, especially those where the authors want to comment the results. 30. Figure 15: What do you mean by the third (and fourth) row and the second row, in the caption? Is it related to the rows of Figure 14? If so it should be clearly stated. 31. P23, l14-15: To be improved 32. p24, l16: this is a model result for just one point in the entire world, hence it is completely irrelevant. Even more when looking at figure 6. Also (see lines 20-22), being in the middle of the 7 outputs doesn't mean it is more skillful. Validation with observed data is recommended.

---

## Author Comment (AC1) · 17 Nov 2020

We are grateful for all the comments. In this discussion forum, we will briefly reply to the comments point by point. The detailed revisions will be shown in our manuscript and the final replies to reviewers.

General comments:

Zhou et al. present a novel and interesting analysis of uncertainties inherent to global hydrodynamic models. Many studies over the past ~5 years have posited that global flood models produce widely divergent results, yet each of them fails to explain or evidence the cause of this divergence. The authors begin to address some of these previous gaps in the literature and have produced some (though limited) conclusions of genuine interest. However, there are fundamental problems with the methods, analysis, and wider interpretation/discussion of their conclusions that require addressing before publication is to be considered in NHESS. Do not be put off by the volume of comments. I do think the work has great value and is something the field desperately requires. To be the truly impactful contribution this work needs to be, I offer the following (I hope, constructive) comments. These broadly relate to:
- Model sensitivity vs. uncertainty analysis

Re: We agree that this paper is discussing the model sensitivity to different climate and model conditions. So, we will reframe the structures and revise the contents in the following aspects.

1.  Adding discussions in the Introduction part how the uncertainty/sensitivity is discussed and addressed in the previous literature.
2.  Revise the introduction in order not to mislead the readers that we are validating our model.
3.  The comparisons with two other methods will be removed from the results since they are not "validation" and they help little to sensitivity analysis.

- A single model with often extremely limited geographic scope

Re: Yes, although we only use one single model, we can analyze model sensitivity to the inputs (i.e., runoff) and other conditions (e.g., fitting distribution, used variables for fitting). If new river models can be added, we can further investigate the flood extent sensitivity to model choice. However, this may need collaboration with different teams.

- Considered uncertainties in the context of 'unconsidered' uncertainties - The relevance of hydrologic variable choice

Re: Yes, we are trying to use different variables in the fitting process because we think different hydrological variables may have different distributions. Although in this paper, only the water level and water storage in the model unit catchment are used, it is still an attempt to investigate a new uncertainty source for the final flood extent mapping.

- The contextual relevance of distribution goodness-of-fit

Re: This is lacking in the current version. We will try to find the relevance of AIC with the results and probably add a few other metrics for an overall evaluation.

- The need for some benchmark data

Re: As the reviewer mentioned, it is not easy to find appropriate benchmark data at the global scale due to lack of enough validation especially in the developing countries. And the two datasets (JRC and GAR) used in this study are not appropriate. Therefore, we will first remove the current section 4.5 (comparison over the lower Mekong region). Then since we decided to focus on the sensitivity analysis at the global scale, we will not use any benchmark data.

\*\*\*\*\*\*\*\*\*\*\*\*\*\*\*\*\*\*\*\*\*\*\*\*\*\*\*\*\*\*\*\*\*\*\*\*\*\*\*\*\*\*\*\*\*\*\*\*\*\*\*\*\*\*\*\*\*\*\*\*

Specific comments:

The introduction is mostly good, but could contain a richer discussion on why unstitch- ing the uncertainties of global flood models is needed and how past studies essentially have failed to do this. There are also many sweeping or incorrect generalisations, that may simply be a function of imprecise English (for which I am sympathetic and under- standing). This is the case throughout the manuscript (e.g. the sentence in line 26-27 p. 2 makes little sense as the same could be said for RFFA; line 3 p. 23 says the studies assess flood risk, when it does not).

Re: Additional literature and new discussions will be added in the introduction. All the text throughout the paper will be checked again and improved by professional English editing service.

The overarching problem in the introduction is that it makes the reader think some quantification of the uncertainties – validation, against observations – is carried out by the authors, and this is not the case. Truly, this analysis is a sensitivity analysis of 1 model. While I appreciate this illustrates the 'uncertainty one should have about conclusions drawn using this model', really it just tells us what the model is sensitive to and by how much. The paper as a whole needs more of a framing as a sensitivity analysis rather than formal uncertainty quantification.

Re: Yes, we have to make our goal more clearly – focusing on the sensitivity rather than validation. The manuscript will be revised accordingly especially the introduction and results.

The methodology, if framed as a sensitivity analysis of CaMa-Flood, appears thorough and fit-for-purpose. In general, justification of the use of a single model and subse- quent analysis in specific regions (even specific grid cells) is needed. How universal are the conclusions in light of these methodological choices? Of course, these are only uncertainties related to the subjective choice of model tests. It is worth stressing that the reported uncertainties make the (of course, incorrect) assumption that terrain, channel bathymetry, human influence, and model parameterisation are certain.

Re: It is difficult to find observational values of the historical-level flood (e.g., 100 years return period). This is why we introduced the results from JRC and GAR to show how CaMa-Flood is compared with modelled (JRC) and more observation-based (GAR) results. (But this part will be removed in the new manuscript because we will more focus on the sensitivity analysis). There are various uncertainties which can lead to diversities in flood extent estimates while in this study we will first discuss a few while assume others are certain.

It is not clear why river depth and water storage are chosen as variables of interest – this needs further explanation, as I can not yet see the significance of doing so. Com- mon application of FFA is to discharge, yet this is not done here. Further discussion of the AIC is needed: what constitutes a 'good' result in this context is not specified. Equally, what is the relevance of this metric in terms of model uncertainty? What is the relevance of a good fitting distribution in the context of the uncertainty in the absolute values themselves? Are the authors saying that a variable with a poor AIC contains no relevant information for FFA? Really, it just shows a suitable distribution has not yet been found. I think section 3.1 fails to recognise the variable of choice is arbitrary and depends on the model used and the question asked. We all know that a 100-year rain- fall$\neq$ 100-year streamflow$\neq$ 100-year economic loss. So frame this strand of analysis in the context of why the variable you choose matters and why this is interesting.

Re: Water storage in the unit-catchment is the prognostic variable in CaMa-Flood. Water level is the diagnostic variable estimated from water storage. With either of these two variables we can estimated the flood extent and the floodplain water depth for any target region. Discharge is the variable frequently used in engineering design. However, with only discharge we cannot estimate the water level since the relation between discharge and water level is not one-to-one consistent due to the loop rating curve.

I can not see evidence that WAK is the best distribution because of it having 5 param- eters. As the authors mentioned, it may just be overfit to the simulation record. The reality is we have no idea which distribution we should extrapolate with – and this is not something the AIC can test.

Re: In this study, we tested six different fitting distributions (e.g., WAK, GEV, etc.). AIC is used to evaluate the fitting performance. However, AIC or other metrics (e.g., RMSE, bias, etc.,) cannot indicate if we should reject the distribution or not. Though, we do not aim to identify the distribution in this study but we mainly want to test how much of the variation can be caused if we use different fitting distributions.

The section 3.3 analysis of runoff is interesting, but the results are stated in such a way that the authors expected the analysis to produce a 'preferable' runoff product. No feature of the analysis performed could identify such a thing. It is not clear why being a runoff product in 'the middle' is the best place to be: it could be that the lowest estimate types are actually best! It is a problem throughout the paper, where a suitable perfor- mance benchmark has not been found. Ensure the results are framed and reported as sensitivities, not as good/bad.

Re: This part will be removed in the new version. Regarding the middle one, we have one pre-described assumption that the users don't know which runoff is the best. In this case, the users tend to use the ensemble rather than a single runoff input. If the system is too heavy to run for all runoffs, it is better to choose the one in the middle. (But anyway, this will be removed from the current manuscript.)

I like the analysis in section 4.1, but I'm not sure why this could not be done for every global grid cell – with normalised results – and presented in the same way. How representative is this grid cell? It may also be interesting here to compare the AIC results to Figure 7c: exploring some of my above comments on why AIC matters more quantitatively (i.e., does high/low AIC [thus, how good the distribution fit is] matter in the context of inundation?)

Re: It is worthy trying to analyze the point uncertainty for all global grid cell. We will think how we can better illustrate/present the results. However, we doubt if it is necessary to work on the normalized results since they will not show the difference caused by biases in the mean value. The current point is selected randomly, so we need to explore whether this is representative at a global scale. We will also explore if AIC can explain part the diversities in this Figure.

As for the rest of section 4, the analysis is good. While I appreciate visualising the globe at this scale is difficult, a lot of the calculations could still easily be done globally. It leaves the reader wondering whether different climatologies and geomorphic settings might have different conclusions. Deltas are difficult to model – particularly for models with poor/no representation of coastal boundaries – and so may have distinct features of uncertainty to other areas. I see no reason for the authors not to report findings elsewhere.

Re: Thanks. It is good to investigate the uncertainties in different climates and locations. This will be added in Section 5 with the global analysis of the uncertainties.

I do not see any value to section 4.5. I have little doubt the CaMa-Flood 100-year map is more accurate than the GAR and JRC maps: it is an uninformative comparison, and certainly not "validation". You only have to look at the stripes of JRC's map in Figure 12b to know that that is not a model you should aspire to resemble! I appreciate finding suitable validation data is difficult, but it is difficult to understand the relevance of the authors' conclusions without some. Perhaps running this analysis in the US or western Europe where high-quality models exist and comparing to those would be a good idea.

Re: We will remove section 4.5. While we may not add analysis in the US or western Europe because observations are still not available. Deleting this part will not affect the uncertainty analysis planned for this study.

Section 5 is strong, but will benefit from drawing on some of the above points. Gen- erally, the manuscript is quite long, and so the impact from section 5 is dampened by unnecessarily long analysis in 4.2-4.4. Throughout the paper, I would ensure each test is a worthwhile inclusion for the conclusions drawn. At present, there are many analyses which offer little additional information which I would consider cutting.

Re: Thanks. Yes, by removing some analysis (e.g., section 4.5) and shortening sections 4.2-4.4, we can explore more on the global maps where the uncertainties are higher and why this happens.

Figures are generally good quality, but most need to be larger. I would change the colour scheme of some figures (e.g. 4-6) where colour scales are used for variables which are not ordinal (no reason to go from blue to red, when the distributions are in no order).

Re: The Figures are prepared in high quality, so it can be enlarged. Regarding the color in Figure 4-6, it is actually near random. But I may change the colorbar which seems that the colors are sequent.

---

## Author Comment (AC2) · 17 Nov 2020

Replies to Referee #2

We are grateful for all the comments. In this discussion forum, we will briefly reply to the comments point by point. The detailed revisions will be shown in our manuscript and the final replies to reviewers.

General comments

This paper is based on large scale hydrologic-hydrodynamic simulations to investigate different sources of uncertainty in flood risk estimation, with the use of flood frequency analysis tools. The chosen topic deserves some interest, though the analysis is based on a specific configuration of a set of available hydrological model output (from the Earth2Observe project) and an in-house hydrodynamic model (CaMa). However, the focus on the global domain makes it of larger interest for a wider community.

- Among the main limitations of the manuscript is the sub-optimal use of the english language, including both terminology, grammar, typos and structure of the sentences, which makes it hard to read and at times hampers the understanding of the content. I strongly suggest to work and improve it with the help of a native speaker.

Re: Thanks for the suggestion. We will seek help from native speakers or English editing company to improve the English.

- Another important comment is related to the general framing of the analysis. In the current version a number of analyses are performed, focusing on different aspects, though in my opinion it lacks a consistent storyline and some reasoning behind why they were made and clear statements about what we learn following their results.

Re: Yes, we do realize that we have included too much analysis from different aspects. In the revised manuscript, we will delete some of them and concentrate on the sensitivity analysis to various model inputs, distributions and the variable selected. Analysis will be conduct from pixel level to basins and to the global scale.

- The manuscript is too long compared to the information content it brings. I suggest shortening following the comments below. A number of figures should be removed, improved or put in the supplement material, for the reasons I explain below in the specific comments. In particular, I'm speaking about Figures 4 and 5 wrt the issues with fitting analytical functions with different degrees of freedom (comment #10), Fig. 6 (comment #18), Fig. 10, 12, and 14 (comments #24, #27, #29)

Re: Thanks, we will shorten the manuscript by considering your comments and comments from the other reviewer. For example, we can combine Figure 4 and Figure 5, delete Figure 6 and Figure 12 since they are not relevant with the sensitivity analysis. We can also delete Figure 10 since it doesn't show enough information. Figure 14 can be improved to show more details in specific zooming regions.

Specific comments

1. p2, l8-9: acronyms should be defined with "full name (acronym)", e.g., Global Runoff Data Centre (GRDC). Same for p3, l5 and l26.

2. p2, l14: Pearson type III

3. p3, l1: suggested "connected" –> "analyzed the relation between ..."

4. p3, l3-5: Sentence not clear. Please rephrase.

5. P4, l3: please define the acronym SAR

6. p4, l10: "various runoff inputs" is too general. Please add details here or a reference to the details included in Sect. 2.2 wrt the inputs used.

7. P4, l13: I suggest adding an introductory sentence here to give more details about the experiment itself, before jumping to the uncertainties to investigate.

8. P4, l14-16: please improve this part. Also, I find the variable names V1_(rivdph) and V2_(sto2dph) not very intuitive. Why not simply calling them depth and storage? Especially sto2dph creates confusion on whether it is a storage or a depth.

9. Table 1: I suggest removing "Various" in the caption.

Re to 1-9: Thanks, we will correct the errors and improve the sentences which have been pointed out above.

10. P5, l12: Note that the Gumbel and the Gamma distributions have 2 parameters. In fact, results in Figure 5 seems to me the natural consequence of fitting a series of points with mathematical functions with different degrees of freedom, where the 5 parameter distribution is able to fit the data more skillfully (though it doesn't mean it will be more skillful in predictive mode for future floods), Then the 3 parameter distributions and the 2-parameter Gamma and Gumbel as the least skillful. One would obtain similar results when fitting the series of data with polynomials of grade 5,3 and 2, because higher grade polynomials can fit better the input data.

Re: Thanks. The degree of freedom is the cause for the diversities of final results using different fitting distributions. We will add this explanation to the revised manuscript.

11. P5, l13: I suggest renaming this section (e.g., "Fitting performance" or similar)

12. p5, l15: calculated

13. p5, l19-20: This should be expressed more clearly. E.g:" Smaller aic denote higher fitting perfor- mance" or similar, which is actually better written in p6, l23-24

14. p6, l24-26: Use active rather than passive form (e.g., "we compare")

Re: We will revise the above and finally the manuscript will be sent for professional English correction.

15. p7, l6-7: Is the normalization the real reason? Also, I suggest giving more details on how to weigh the aic values. What is the optimum? What are normally considered good or bad values? It is not intuitive for those who have never used it.

Re: We will give more information of AIC in the Methods. AIC is especially suitable for evaluating model performance with a narrow value range (e.g., 0-1 in this study), because it enlarges the difference by logarithm.

16. P8, l8: "The later peak" – > "the latter"

Re: ok.

17. Figure 3: Interesting to see how the pdfs of gamma and gumbel have similar peaks to the other distributions only for the storage, but not for the river depth. Indeed it is clearly fisible also in Fig. 3c. Would be interesting to investigate and motivate the reasons. Now it is only mentioned but no justification is given.

Re: Thanks. This could be an interesting point. We will explore the reasons and add it to the revised manuscript.

18. Figure 6: How does this analysis relate to the FFA and to the rest of the paper in general? I'm not sure of the value of these maps, given the little information the readers have on the 7 runoff inputs, and also because there is no clear patter identified. Perhaps the main information one can obtain is that anu and univu tends to be on the lower side, while cnrs and univk on the higher side. Yet, this doesn't say anything about the skills of these estimates, which would imply validation with gauge data at a number of stations.

Re: We decided to remove this subsection (and Figure 6) because it is not very relevant to the rest of the paper.

19. P13, l2: after – > downstream

Re: ok.

20. Note that Figure 8 is referenced before Figure 7

Re: Thanks, we will correct this in the revised manuscript.

21. Sect 4.1 refers to return periods in Fig.7, hence in Fig.7 I advise to show return periods in place of frequencies. In any case, to be correct you should refer to those as annual frequencies of occurrence, to avoid confusion. Also, in Figure 7c, why not all distributions are shown?

Re: Thanks. We will revise the x-axis label and ticks. We didn't show the results from Gamma and Gumbel because the fitting performance for these two fitting distributions are the lowest among all the six distributions. We will add them in the revised manuscript.

22. P14, l2: please give some details and possibly a reference on the downscaling procedure.

Re: Ok, the downscaling procedure will be added to the Methods.

23. P14, l3-6: To aid the assessment of water depths I suggest showing in Fig.8 a map or contour of the permanent water bodies. Clearly it is normal to have higher water depths in rivers and lakes, compared to areas normally dry. Also, I cannot find information about the terrain model, in particular whether it represents the river bed or some reference water level. This is important for this analysis.

Re: Thanks. We use Multi-Error-Removed Improved-Terrain DEM (MERIT DEM) as the terrain model. We will also prepare a map for permanent water bodies and added to Figure 8.

24. Figure 10: results shown in this figure are rather obvious. I suggest removing this figure as it brings little information. Over large inundation depths it is normal to have good agreement on whether there's inundation or not, as having poor agreement would mean huge differences in the results of the model used (hence very poor skills for some models).

Re: Thanks, this Figure will be removed.

25. P17, l11: return periods should not be expressed as percentage

Re: Thanks. We will correct it.

26. p18, l10 and Figure 12: Is this the mean inundation of the 7 models? Clarify

Re: They are the mean values among all different experiments, with different runoff inputs, fitting distributions and two selected variables. We will clarify this in the revised manuscript.

27. I find the analysis in Figure 12 of limited use, being a qualitative visual comparison with two other publicly available maps, but also resulting from modeling exercise with limited calibration. Similarly, the comments in p19, l14-18 are partly speculative. More rigorous validation with observed flooded areas would give much more strength to the paper.

Re: Thanks, the comparison of CaMa-Flood result to the other two sources (Figure 12 and subsection 4.5) will be removed in the revised manuscript.

28. P 21,l6: for flood impact assessment it is more interesting to know (even smaller) inundation depths in areas where people live or where economic assets are, rather than the inundation in the main channels, which has fewer fields of application.

Re: Yes, the population exposure or GDP exposure to floods is one of key interests in flood damage assessment. We will think about whether we add these assessments in our updates.

29. Figure 14 is unreadable and of limited use in the present form. It is impossible to get enough spatial details of a global inundation map at such small scales. Furthermore, the left and right column are almost indistinguishable. I suggest removing this figure and rather put it in the supplement, together with a number of inset panels zooming into some areas, especially those where the authors want to comment the results.

Re: Thanks, we will think about how to better present the results with this global Figure. And zooming panels can be added if we want to discuss on some regions.

30. Figure 15: What do you mean by the third (and fourth) row and the second row, in the caption? Is it related to the rows of Figure 14? If so it should be clearly stated.

Re: Yes. The captions links Figure 14. We will clarify this in the revised manuscript.

31. P23, l14-15: To be improved

Re: OK.

32. p24, l16: this is a model result for just one point in the entire world, hence it is completely irrelevant. Even more when looking at figure 6. Also (see lines 20-22), being in the middle of the 7 outputs doesn't mean it is more skillful. Validation with observed data is recommended.

Re: Yes, we also mentioned that the point analysis for only one point is not relevant. As reviewer #1 mentioned, we can try to analyze the point values but for all global grids. This will help to find the general results for the global scale. We can validate our model discharge with GRDC observations. This will be added to the supporting information.

---

## Author Response (AR1)

Dear editor and reviewers,

We are grateful for all your comments. In this response document, we replied to the comments point by point. The detailed revisions are shown in our manuscript.

Replies to Referee #1

General comments:

Zhou et al. present a novel and interesting analysis of uncertainties inherent to global hydrodynamic models. Many studies over the past ~5 years have posited that global flood models produce widely divergent results, yet each of them fails to explain or evidence the cause of this divergence. The authors begin to address some of these previous gaps in the literature and have produced some (though limited) conclusions of genuine interest. However, there are fundamental problems with the methods, analysis, and wider interpretation/discussion of their conclusions that require addressing before publication is to be considered in NHESS. Do not be put off by the volume of comments. I do think the work has great value and is something the field desperately requires. To be the truly impactful contribution this work needs to be, I offer the following (I hope, constructive) comments. These broadly relate to:
- Model sensitivity vs. uncertainty analysis

Re: We agree that the contribution of this paper should be the model sensitivity as the reviewers suggested. So, we reframed the structures and revised the manuscript in the following aspects.

1. Added discussions in the Introduction how the uncertainty/sensitivity is discussed and addressed in the previous literature.
2. Emphasized in the introduction that we focused on uncertainty/sensitivity analysis rather than validation of the flood estimation.
3. The comparisons with two other flood hazard maps are removed from the results since they are not "validation" and they help little to uncertainty/sensitivity analysis.
4. We added uncertainties/sensitivity analysis on the population exposure and economic exposure for different floods.

- A single model with often extremely limited geographic scope

Re: Agree. The flood method is one of the uncertainties can lead to deviations in the estimation. However, we are not able to run different routing models with different runoff inputs. This may need more collaborations. Although we only use one single model, we can analyze model sensitivity to the inputs (i.e., runoff) and other conditions (e.g., fitting distribution, used variables for fitting). If new river models are added, we would further investigate the flood extent sensitivity to model choice. The limitation is discussed in the Discussions.

Trigg et al. (2016), Bernhofen et al. (2018) and Aerts et al. (2020) used 8 different models with different forcing, hydrological model, routing models. Although their results showed

deviations, they are not able to attribute the contribution a single process (e.g., routing) to the final uncertainty. (This has been added in the Introduction)

Trigg, M. A., Birch, C. E., Neal, J. C., Bates, P. D., Smith, A., Sampson, C. C., Yamazaki, D., Hirabayashi, Y., Pappenberger, F., Dutra, E., Ward, P. J., Winsemius, H. C., Salamon, P., Dottori, F., Rudari, R., Kappes, M. S., Simpson, A. L., Hadzilacos, G., and Fewtrell, T. J.: The credibility challenge for global fluvial flood risk analysis, Environmental Research Letters, 11, https://doi.org/10.1088/1748- 9326/11/9/094014, 2016.

Bernhofen, M. V., Whyman, C., Trigg, M. A., Sleigh, P. A., Smith, A. M., Sampson, C. C., Yamazaki, D., Ward, P. J., Rudari, R., Pappenberger, F., Dottori, F., Salamon, P., and Winsemius, H. C.: A first collective validation of global fluvial flood models for major floods in Nigeria and Mozambique, Environmental Research Letters, 13, https://doi.org/10.1088/1748-9326/aae014, 2018.

Aerts, J. P. M., Uhlemann-Elmer, S., Eilander, D., and Ward, P. J.: Comparison of estimates of global flood models for flood hazard and exposed gross domestic product: a China case study, Natural Hazards and Earth System Sciences, 20, 3245–3260, https://doi.org/10.5194/nhess-20-3245-2020, 2020.

- Considered uncertainties in the context of 'unconsidered' uncertainties - The relevance of hydrologic variable choice

Re: Yes, we are using different variables in the fitting process (i.e., water level, water storage) because it is not linear between water level and water storage. They must have different distributions. Although in this paper, only the water level and water storage in the model unit catchment are used, it is still an attempt to investigate a new uncertainty source for the final flood extent mapping. Discharge is frequently used in engineering projects, while it is not used in our analysis because discharge and water level is not one-to-one consistent due to the loop rating curve. While with either of river water depth or the water storage we can estimated the flood extent and the floodplain water depth for any target region using CaMa-Flood. (This has been reflected in section 2.1)

- The contextual relevance of distribution goodness-of-fit

Re: In the new manuscript, we conclude that the goodness-of-fit is likely to be poor in dry climate or mountainous areas. These regions are where the accumulative river discharge is small. The magnitude of water level (or river storage) is highly depended on single precipitation events, leading to an unstable relation between the high floods in different years. (This has been reflected in section 3.1)

- The need for some benchmark data

Re: As the reviewer mentioned, it is not easy to find appropriate benchmark data at the global scale due to lack of enough validation data especially in the developing countries. The two datasets (JRC and GAR) used in this study are also not appropriate because they are still replying on models. However, since we decided to focus on the sensitivity analysis at the

global scale, we first removed the current section 4.5 (comparison over the lower Mekong region) and we will not use any benchmark data (including any statistics or remote sensing images).

\*\*\*\*\*\*\*\*\*\*\*\*\*\*\*\*\*\*\*\*\*\*\*\*\*\*\*\*\*\*\*\*\*\*\*\*\*\*\*\*\*\*\*\*\*\*\*\*\*\*\*\*\*\*\*\*\*\*\*\*

Specific comments:

The introduction is mostly good, but could contain a richer discussion on why unstitching the uncertainties of global flood models is needed and how past studies essentially have failed to do this. There are also many sweeping or incorrect generalisations, that may simply be a function of imprecise English (for which I am sympathetic and under- standing). This is the case throughout the manuscript (e.g. the sentence in line 26-27 p. 2 makes little sense as the same could be said for RFFA; line 3 p. 23 says the studies assess flood risk, when it does not).

Re: We have rewritten the Introduction. Additional literature and new discussions on the sensitivity analysis are provided.

To avoid English errors, the whole text has been checked with professional English editing service (https://www.proof-reading-service.com). A certificate is provided as a supporting material.

The overarching problem in the introduction is that it makes the reader think some quantification of the uncertainties – validation, against observations – is carried out by the authors, and this is not the case. Truly, this analysis is a sensitivity analysis of 1 model. While I appreciate this illustrates the 'uncertainty one should have about conclusions drawn using this model', really it just tells us what the model is sensitive to and by how much. The paper as a whole needs more of a framing as a sensitivity analysis rather than formal uncertainty quantification.

Re: Yes, we have restructured the whole paper and now this study is concentrated on the sensitivity analysis to various influential factors, rather than quantifying differences/biases compared to observations. The introduction has been rewritten as well.

The methodology, if framed as a sensitivity analysis of CaMa-Flood, appears thorough and fit-for-purpose. In general, justification of the use of a single model and subsequent analysis in specific regions (even specific grid cells) is needed. How universal are the conclusions in light of these methodological choices? Of course, these are only uncertainties related to the subjective choice of model tests. It is worth stressing that the reported uncertainties make the (of course, incorrect) assumption that terrain, channel bathymetry, human influence, and model parameterisation are certain.

Re: The compassions with other products have been removed.

There are indeed many uncertainty sources which are not studied in this paper. So, we assume others are certain and we had in our Discussions about how other uncertainties sources have a potential impact on our results and how important is the uncertainty in other studies..

a. It is not clear why river depth and water storage are chosen as variables of interest – this needs further explanation, as I can not yet see the significance of doing so. Common application of FFA is to discharge, yet this is not done here. b.Further discussion of the AIC is needed: what constitutes a 'good' result in this context is not specified. Equally, what is the relevance of this metric in terms of model uncertainty? What is the relevance of a good fitting distribution in the context of the uncertainty in the absolute values themselves? c.Are the authors saying that a variable with a poor AIC contains no relevant information for FFA? Really, it just shows a suitable distribution has not yet been found. I think section 3.1 fails to recognise the variable of choice is arbitrary and depends on the model used and the question asked. We all know that a 100-year rain-fall≠ 100-year streamflow≠ 100-year economic loss. So frame this strand of analysis in the context of why the variable you choose matters and why this is interesting.

Re: a. Water storage in the unit-catchment is the prognostic variable in CaMa-Flood. Water level is the diagnostic variable estimated from water storage. With either of these two variables we can estimated the flood extent and the floodplain water depth for any target region. Discharge is the variable frequently used in engineering design. However, with only discharge we cannot estimate the water level since the relation between discharge and water level is not one-to-one consistent due to the loop rating curve. (This has been reflected in 2.1 Experiment design).

b. Instead of defining the best/good fitting result with aic, we decided to compare the sensitivity of the aic to different experiment. The aic itself is not able to show the model uncertainty, but when we compare the aic for all experiments, we can conclude that aic is sensitive to the selection of the fitting functions, and then the variables. It is less sensitive to the runoff inputs. (additional explanation of the aic is added in section 2.3, the results are reflected in section 3.1)

c. No, as the reviewer mentioned, we haven't found a best fitting distribution for any variable that is applicable for all the places. But we can still use the uncertainty information from experiments with different variables. Regarding other variables, because we are working on floods, 100-year rainfall is not applicable. 100-year streamflow is discussed in the reply to this question a. 100-year economic loss could be more complex. But we added one subsection in the result to discuss what are the population and economic exposure to the floods (see section 3.3.3).

I can not see evidence that WAK is the best distribution because of it having 5 parameters. As the authors mentioned, it may just be overfit to the simulation record. The reality is we have no idea which distribution we should extrapolate with – and this is not something the AIC can test.

Re: Yes, we agree. Therefore, in the new manuscript, the aic is used for testing the sensitivity of fitting performance by comparing aic for various experiments.

The section 3.3 analysis of runoff is interesting, but the results are stated in such a way that the authors expected the analysis to produce a 'preferable' runoff product. No feature of the analysis performed could identify such a thing. It is not clear why being a runoff product in 'the middle' is the best place to be: it could be that the lowest estimate types are actually best! It is a problem throughout the paper, where a suitable perfor- mance benchmark has not been found. Ensure the results are framed and reported as sensitivities, not as good/bad.

Re: This part has been removed in the new manuscript. Regarding the "middle" one, we have one pre-assumption that the users don't know which runoff should be the best. In this case, the users tend to use the ensemble mean rather than a single runoff input. If the system is too heavy to run for all runoffs, it is better to choose the one in the middle. (But anyway, this has been removed from the previous manuscript.)

I like the analysis in section 4.1, but I'm not sure why this could not be done for every global grid cell – with normalised results – and presented in the same way. How representative is this grid cell? It may also be interesting here to compare the AIC results to Figure 7c: exploring some of my above comments on why AIC matters more quantitatively (i.e., does high/low AIC [thus, how good the distribution fit is] matter in the context of inundation?)

Re: Probably, it is not necessary to work on the normalized results since they will neglect the variations caused by the mean values. Conducting the point analysis for all the global grids might not be realistic with regard to the necessity or the difficulty for visualization. Instead, we selected five more river basins (e.g., Amazon, Yangtze, Mississippi, Lena and Nile), for each river basin we investigated the results at one specific GRDC gauge near the river outlet.

The differences of the fitting performance are mainly due to the degree of freedom of each fitting distribution as the WAK has five parameters, GAM and GUM have two parameters while the others have three. With higher degree of freedom, the fitting performance will be better. Regarding the spatial pattern, we found that the aic is higher (indicating poorer fitting performance) in dry zones and mountainous regions. The accumulative river discharge over those regions is small. The magnitude is thus highly depended on single precipitation events, leading to an unstable relation between the high floods in different years. (This has been reflected in section 3.1).

As for the rest of section 4, the analysis is good. While I appreciate visualising the globe at this scale is difficult, a lot of the calculations could still easily be done globally. It leaves the reader wondering whether different climatologies and geomorphic settings might have different conclusions. Deltas are difficult to model – particularly for models with poor/no representation of coastal boundaries – and so may have distinct features of uncertainty to other areas. I see no reason for the authors not to report findings elsewhere.

Re: Thank you very much for this suggestion. In the new manuscript, the analysis over the global maps has been added as current section 3.2. For short, the regions with high coefficient of variation (Cv, ratio of standard deviation to the mean) are likely to be the dry zones (e.g., Sahara, Center Australia, Center Asia) and the originating river basins in mountainous regions (e.g., the Rocky Mountain, the Andes, Tibet Plateau). We didn't find large variations over the deltas, this might because we haven't introduced other flood models. With the same flood model, the deviation among different inputs/fitting functions can be small over the deltas.

Visualizing the global results is difficult, we therefore provide detailed maps for specific river basins. The results for the lower Mekong River Basin are presented in the main text, while results for five other river basins (i.e., Amazon, Yangtze, Mississippi, Lena, Nile) are provided in the supporting material.

I do not see any value to section 4.5. I have little doubt the CaMa-Flood 100-year map is more accurate than the GAR and JRC maps: it is an uninformative comparison, and certainly

not "validation". You only have to look at the stripes of JRC's map in Figure 12b to know that that is not a model you should aspire to resemble! I appreciate finding suitable validation data is difficult, but it is difficult to understand the relevance of the authors' conclusions without some. Perhaps running this analysis in the US or western Europe where high-quality models exist and comparing to those would be a good idea.

Re: We have removed this section. And we didn't add analysis in the US or western Europe because observations are still not available. Excluding this part will not affect the current scope of this study.

Section 5 is strong, but will benefit from drawing on some of the above points. Generally, the manuscript is quite long, and so the impact from section 5 is dampened by unnecessarily long analysis in 4.2-4.4. Throughout the paper, I would ensure each test is a worthwhile inclusion for the conclusions drawn. At present, there are many analyses which offer little additional information which I would consider cutting.

Re: Thanks. Yes, by removing some analysis (e.g., section 4.5) and shortening sections 4.2-4.4, we can explore more on the global maps where the uncertainties are higher and why this happens. The detailed analysis can be found as section 3.2.

Figures are generally good quality, but most need to be larger. I would change the colour scheme of some figures (e.g. 4-6) where colour scales are used for variables which are not ordinal (no reason to go from blue to red, when the distributions are in no order).

Re: The Figures are prepared with high quality, so it can be displayed in a large size. Regarding the color in Figure 4-6, it is actually near random. But I may change the colorbar which seems that the colors are sequent (but the mentioned figures are removed from the text.)

General comments

This paper is based on large scale hydrologic-hydrodynamic simulations to investigate different sources of uncertainty in flood risk estimation, with the use of flood frequency analysis tools. The chosen topic deserves some interest, though the analysis is based on a specific configuration of a set of available hydrological model output (from the Earth2Observe project) and an in-house hydrodynamic model (CaMa). However, the focus on the global domain makes it of larger interest for a q      wider community.

- Among the main limitations of the manuscript is the sub-optimal use of the english language, including both terminology, grammar, typos and structure of the sentences, which makes it hard to read and at times hampers the understanding of the content. I strongly suggest to work and improve it with the help of a native speaker.

Re: Thanks for the suggestion. The submitted revision has been checked with professional English editing service (https://www.proof-reading-service.com). A certificate is provided as a supporting material.

- Another important comment is related to the general framing of the analysis. In the current version a number of analyses are performed, focusing on different aspects, though in my opinion it lacks a consistent storyline and some reasoning behind why they were made and clear statements about what we learn following their results.

Re: Yes, we realized that we have included too much analysis from different aspects. In the revised manuscript, we deleted some of them and concentrated on the sensitivity analysis to various model inputs, fitting functions and the variable selected. Analysis will be conducted from the global scale to basin scale with emphasizing point analysis. The uncertainties in the inundation area and the potential impact on population and economy is also discussed with the uncertainties.

- The manuscript is too long compared to the information content it brings. I suggest shortening following the comments below. A number of figures should be removed, improved or put in the supplement material, for the reasons I explain below in the specific comments. In particular, I'm speaking about Figures 4 and 5 wrt the issues with fitting analytical functions with different degrees of freedom (comment #10), Fig. 6 (comment #18), Fig. 10, 12, and 14 (comments #24, #27, #29)

Re: Thanks, we have shortened the manuscript by considering your comments and comments from reviewer #1. For example, we deleted Figure 4 to Figure 6 because they are not relevant to the sensitivity analysis. We deleted Figure 9 and 10 since they bring little information (as #24). Figure 12 and 13 are removed as we will not discuss the validation in this study. Figure 14 is reorganized so that it can be better discussed. New Analysis on the sensitivity over many other river basins are added in the support materials to support our analysis. Analyses on the population and economic exposure to the floods are added.

Specific comments

1. p2, l8-9: acronyms should be defined with "full name (acronym)", e.g., Global Runoff Data Centre (GRDC). Same for p3, l5 and l26.

Re: Thanks, the same errors are fixed in the text.

2. p2, l14: Pearson type III

Re: Revised.

3. p3, l1: suggested "connected" –> "analyzed the relation between ..."

Re: Revised.

4. p3, l3-5: Sentence not clear. Please rephrase.

Re: The Introduction has been almost rewritten.

5. P4, l3: please define the acronym SAR

Re: Synthetic-aperture radar (SAR). Added in the manuscript.

6. p4, l10: "various runoff inputs" is too general. Please add details here or a reference to the details included in Sect. 2.2 wrt the inputs used.

Re: Yes, the various runoff inputs are listed in section 2.1 now. We added a reference to the previous section to explain the "various runoff inputs".

7. P4, l13: I suggest adding an introductory sentence here to give more details about the experiment itself, before jumping to the uncertainties to investigate.

Re: We reorganized the method part. One new paragraph is added before introducing the experiments. (This is reflected in first paragraph in the section 2.1)

8. P4, l14-16: please improve this part. Also, I find the variable names V1_(rivdph) and V2_(sto2dph) not very intuitive. Why not simply calling them depth and storage? Especially sto2dph creates confusion on whether it is a storage or a depth.

Re: Thanks. As suggested, in the new manuscript, we use river water depth and water storage instead of "rivdph" or "sto2dph". Explanations of the differences between using water level and water storage, as well as the reason for not using discharge are added in the manuscript (see the second paragraph in section 2.1). The explanations also can be found in replies to the fourth question of Reviewer #1.

9. Table 1: I suggest removing "Various" in the caption.

Re: Revised.

10. P5, l12: Note that the Gumbel and the Gamma distributions have 2 parameters. In fact, results in Figure 5 seems to me the natural consequence of fitting a series of points with mathematical functions with different degrees of freedom, where the 5 parameter distribution

is able to fit the data more skillfully (though it doesn't mean it will be more skillful in predictive mode for future floods), Then the 3 parameter distributions and the 2-parameter Gamma and Gumbel as the least skillful. One would obtain similar results when fitting the series of data with polynomials of grade 5,3 and 2, because higher grade polynomials can fit better the input data.

Re: Thanks. The degree of freedom is the cause for the diversities of final results using different fitting distributions. We have added this explanation to the revised manuscript in section 3.1.

11. P5, l13: I suggest renaming this section (e.g., "Fitting performance" or similar)

Re: Revised. The subtitles for other sections are also checked.

12. p5, l15: calculated

Re: revised.

13. p5, l19-20: This should be expressed more clearly. E.g:" Smaller aic denote higher fitting performance" or similar, which is actually better written in p6, l23-24

Re: we also added just after the equation that the smaller aic denotes higher fitting performance in section 2.3.

14. p6, l24-26: Use active rather than passive form (e.g., "we compare")

Re: We revised the above and sentences in similar locations. Before resubmitting the revision, the manuscript has been sent for professional English correction to minimize such problems.

15. p7, l6-7: Is the normalization the real reason? Also, I suggest giving more details on how to weigh the aic values. What is the optimum? What are normally considered good or bad values? It is not intuitive for those who have never used it.

Re: Although, there are various performance metrics to measure the goodness-of-fit, the aic is better in our study because it will enlarge the small difference between samples and estimations. We only have 35 samples and these are sorted, therefore the fitting performance should be very high and fitting results will not have large differences. Then optimum of aic is negative infinity. Same as RMSE, the lower AIC, the performance is better.

16. P8, l8: "The later peak" – > "the latter"

Re: Revised.

17. Figure 3: Interesting to see how the pdfs of gamma and gumbel have similar peaks to the other distributions only for the storage, but not for the river depth. Indeed it is clearly fisible also in Fig. 3c. Would be interesting to investigate and motivate the reasons. Now it is only mentioned but no justification is given.

Re: Thanks. Investigating the difference in fitting the water level and the storage is interesting. As shown in the following illustration, in CaMa-Flood the water level is calculated by allocating the water storage to the river channel and floodplain from the bottom to the top. In the channel, the relation between water level and water storage is linear because we assume the river profile as rectangle, while it changes to nonlinear in the floodplains. So, if the maximum water level for the different years locates both in the river channel and the floodplain, fitting the water level becomes more difficult especially for GAM and GUM since they only have 2-parameters. While, because the storage is not affected by the channel shape, fitting the water storage with different fitting functions will not make large differences. (This has been reflected in the section 3.1).

[Figure]

18. Figure 6: How does this analysis relate to the FFA and to the rest of the paper in general? I'm not sure of the value of these maps, given the little information the readers have on the 7 runoff inputs, and also because there is no clear patter identified. Perhaps the main information one can obtain is that anu and univu tends to be on the lower side, while cnrs and univk on the higher side. Yet, this doesn't say anything about the skills of these estimates, which would imply validation with gauge data at a number of stations.

Re: We decided to remove this subsection (and Figure 4-6) because it is not very relevant to the rest of the paper now.

19. P13, l2: after – > downstream

Re: revised.

20. Note that Figure 8 is referenced before Figure 7

Re: Thanks, we have re-ordered the texts and the figures, so this problem is fixed.

21. Sect 4.1 refers to return periods in Fig.7, hence in Fig.7 I advise to show return periods in place of frequencies. In any case, to be correct you should refer to those as annual frequencies of occurrence, to avoid confusion. Also, in Figure 7c, why not all distributions are shown?

Re: Thanks. We have revised the x-axis label and ticks to return period. We didn't show the results from Gamma and Gumbel because the fitting performance for these two fitting

distributions are the lowest among all the six distributions. But we have added them in the new figures in revised manuscript.

22. P14, l2: please give some details and possibly a reference on the downscaling procedure.

Re: Ok, the downscaling procedure is added to the Methods. A reference (Zhou et al., 2020) is also added for details about the downscaling method.

Zhou, X., Prigent, C., and Yamazaki, D.: Reasonable agreements and mismatches between land-surface-water-area estimates based on a global river model and Landsat data, Earth and Space Science Open Archive, p. 31, https://doi.org/10.1002/essoar.10504917.1, https://doi.org/10.1002/essoar.10504917.1, 2020.

23. P14, l3-6: To aid the assessment of water depths I suggest showing in Fig.8 a map or contour of the permanent water bodies. Clearly it is normal to have higher water depths in rivers and lakes, compared to areas normally dry. Also, I cannot find information about the terrain model, in particular whether it represents the river bed or some reference water level. This is important for this analysis.

Re: Thanks. We use Multi-Error-Removed Improved-Terrain DEM (MERIT DEM) as the terrain model (this is added in section 2.4 and the figure caption). Sorry that we decided not to add the permanent water bodies because of the following two reasons. First, the accuracy of the estimated permanent water still needs validation, while it is not appropriate to use permanent water from satellite (e.g., Landsat, Pekel et al., 2016) since it may not match the model estimation because limitation of satellites as well. Second, the permanent water for the deltas can be very small compared to the flooded area at the 100-yr return period. It may not help a lot to our analysis.

24. Figure 10: results shown in this figure are rather obvious. I suggest removing this figure as it brings little information. Over large inundation depths it is normal to have good agreement on whether there's inundation or not, as having poor agreement would mean huge differences in the results of the model used (hence very poor skills for some models).

Re: Thanks, this Figure and relevant texts have been deleted in the new manuscript.

25. P17, l11: return periods should not be expressed as percentage

Re: Thanks. We have corrected it.

26. p18, l10 and Figure 12: Is this the mean inundation of the 7 models? Clarify

Re: Yes, the map represents the mean values among all different experiments, with different runoff inputs, fitting distributions and two selected variables. However, this has been removed from the manuscript.

27. I find the analysis in Figure 12 of limited use, being a qualitative visual comparison with two other publicly available maps, but also resulting from modeling exercise with limited calibration. Similarly, the comments in p19, l14-18 are partly speculative. More rigorous validation with observed flooded areas would give much more strength to the paper.

Re: Thanks, the comparison of CaMa-Flood result to the other two sources (Figure 12 and subsection 4.5) has been removed in the revised manuscript.

We are working on the calibration/validation of the CaMa-Flood with observed flooded areas, but that work is out of scope of this study. So, we decided not to mention the validation in this paper.

28. P 21,l6: for flood impact assessment it is more interesting to know (even smaller) inundation depths in areas where people live or where economic assets are, rather than the inundation in the main channels, which has fewer fields of application.

Re: Thanks for the comments! The population exposure or GDP exposure to floods is one of key interests in flood damage assessment. In the new manuscript, we added analysis about the population exposure and economic exposure to floods in different continents. There are some interesting conclusions.

1. Asia will suffer the largest flood extent, population exposure and economic exposure among all the continents. While the proportion of the inundation area in North America is the largest, although the population and economic exposure ratio are very low in North America.

2. Africa has a lower ratio of inundation area, but relative high exposure of population and economy. This indicates that Africa replies very heavily the water and flood will cause significant damages.

3. The uncertainties in Africa is the largest, indicating that models are not consistent in Africa. This can be caused by the complexity of the topography and climate zones in Africa.

Detailed discussions can be found in the section 3.3.3 in the manuscript.

29. Figure 14 is unreadable and of limited use in the present form. It is impossible to get enough spatial details of a global inundation map at such small scales. Furthermore, the left and right column are almost indistinguishable. I suggest removing this figure and rather put it in the supplement, together with a number of inset panels zooming into some areas, especially those where the authors want to comment the results.

Re: The result for floods at 1-in-20 years return period has been removed from Figure 14. In order to see more details about the inundation, we added analysis on five other river basins (e.g., Amazon, Yangtze, Mississippi, Lena and Nile). Readers can find the details if checking the zoomed map for specific river basins. The maps are added in the supporting materials as Figure S1-S5.

30. Figure 15: What do you mean by the third (and fourth) row and the second row, in the caption? Is it related to the rows of Figure 14? If so it should be clearly stated.

Re: Yes. The captions links Figure 14. But in the new manuscript, the Figure 15 is deleted.

31. P23, l14-15: To be improved

Re: We have already rewritten the discussions in the new manuscript.

32. p24, l16: this is a model result for just one point in the entire world, hence it is completely irrelevant. Even more when looking at figure 6. Also (see lines 20-22), being in the middle of the 7 outputs doesn't mean it is more skillful. Validation with observed data is recommended.

Re: The point analysis helps analyze the uncertainties at different return period. It is not applicable to check all the points in the world, thus we selected one representative point (GRDC gauge near the basin outlet) for each river basin to be investigated. Beyond the point analysis, we also analyzed the inundation area, showing similar uncertainty changes for different return periods as the point results. We agree with the reviewer that one single model for one point is sufficient to the judgement of the best model. Therefore, in the new manuscript, we avoid justifying which model/fitting function is the best. But we focus on the uncertainty/sensitivity rather than the accuracy of the model. Validation of CaMa-Flood against river discharge has been provided in Hirabayashi et al., 2013. Model validation with observed data (e.g., inundation area, water altimetry) is now ongoing. However, we think the validation is out of the scope of this study, since we would like to discuss more on the sensitivities from different influential factors.

---

## Author Response (AR2)

Replies to reviewer #1

The revised article is substantially different from the previous submission, and addresses most of my initial comments. Also the quality of the english language has improved, with only some typos left. Also please check the ending "s" for third person singular form. I recommend that the article is published provided that the comments below are adequately addressed.

P1 l21: It is not observable over large areas, but it is observable over specific river reaches.

Re: Thanks, to be accurate, we revised the sentence as

"However, FHM is a theoretical map of a global-identical reoccurrence (e.g., 1-in-100 year return period), and thus it is **difficult to be observed, especially at a large scale**."

P9 l 18-19: Here I would call it "the ranking" rather than the relative magnitude

Re: "**The relative magnitude**" is now replaced by "**the ranking**".

Figure 5: Why are VARIABLES, RUNOFFS and FUNCTIONS uppercase? Also, fix the typo on the y axis, it should be "Proportion". (Same for Figure S1-S5). Also, increase the visibility of the yellow cross.

Re: We intended to emphasize the different groups. Now the uppercases are revised and the figure titles are presented in just a natural way. The typo "proportion" has been fixed in all figures and we have enlarged the yellow cross in the figure.

P15, l3: e.g, "should be taken with caution".

Re: Thanks, this sentence is revised as

"This inundation will lead to migration and economic losses, and the impact **should be taken with caution** because of the uncertainties in inundation estimations."

P18 l2: I don't understand why "calibration will ruin the designed sensitivity test with different runoffs". That shouldn't happen if you use the same calibrated setup for all runs to compare.

Re: Sorry, the words lead to some confusions. We meant that we cannot re-calibrate our model for each specific runoff. For instance, if we apply one new runoff and we target to minimize the bias in discharge peaks by tuning model parameters, the final results related to the floods (e.g., flood water depth, flood inundation) will be useless for the sensitivity test. As the reviewer said, we have optimized the model parameters and validated our results driven by another given runoff (Lin et al., 2020) against GRDC records globally (although the results are not shown in this

paper). Previous publications (Hirabayashi et al., 2013, Yamazaki et al., 2011) have also validated that CaMa-Flood is able to be used for flood assessment globally.

In the new submission, we revised the current sentence as:

> *"**However**, in this study, CaMa-Flood **was not calibrated against observations for each specific runoff** because **additional** calibration will ruin the designed sensitivity test."*

Lin, P., Pan, M., Beck, H. E., Yang, Y., Yamazaki, D., Frasson, R., David, C. H., Durand, M., Pavelsky, T. M., Allen, G. H., Gleason, C. J., & Wood, E. F. (2019). Global Reconstruction of Naturalized River Flows at 2.94 Million Reaches. *Water Resources Research*, *55*(8), 6499–6516. https://doi.org/10.1029/2019WR025287

Hirabayashi, Y., Mahendran, R., Koirala, S., Konoshima, L., Yamazaki, D., Watanabe, S., Kim, H., & Kanae, S. (2013). Global flood risk under climate change. *Nature Climate Change*, *3*(9), 816–821. https://doi.org/10.1038/nclimate1911

Yamazaki, D., Kanae, S., Kim, H., & Oki, T. (2011). A physically based description of floodplain inundation dynamics in a global river routing model. *Water Resources Research*, *47*(4), 1–21. https://doi.org/10.1029/2010WR009726

Replies to reviewer #2

The authors are to be commended for the improvements to their manuscript and the interesting results they present. A number of issues still remain outstanding, however, which require addressing before publication is to be considered:

1. [a]Language remains an issue in this manuscript. I appreciate the efforts of the authors, but the language as a whole remains difficult to follow and poor translation has led to some questionable commentary in some places. The copyeditors of the journal will be able to help with some of the more cosmetic errors, but the substantive meaning of what has been said needs to be checked by the authors. In particular, [b]I'm not sure diagnostic/prognostic variables is a useful descriptor, and [c]in the abstract it states that runoff is 80% of total uncertainties -- please make clear this is the total of considered uncertainties, given an analysis of very large uncertainties (e.g. terrain data errors, river bathymetry) is not considered.

Re: [a] Thanks, we checked throughout the manuscript and corrected/refined the texts we found. We think this will help improve the readability of the paper.

[b] The prognostic variable (water storage) and diagnostic variable (water level) are specific for the river model (CaMa-Flood). Although we treat the two variables in a same way in sensitivity analysis in this paper, they mean differently in the CaMa-Flood as the primary causes to their variations are different as well. For instance, the variations as well as the uncertainties in water storage will be mainly caused by the water flux, while the bias of water level can also be significantly affected due to bias in the river bathymetry.

In the section 2.2 Global river routing model (CaMa-Flood), we added explanation:

> "Therefore, the estimation of water level will additional contain uncertainties in river bathymetry and topography, while uncertainties in the water storage are dominant by the water flux."

[c] Yes, thanks the reviewer's comment. The quantification is only among what have been investigated in this paper (i.e., variables, runoffs and functions).

> "Our results show that deviation in the runoff inputs is the most influential source of uncertainties in the estimated flooded water depth and inundation area, contributing more than 80% of the total uncertainties ***investigated in the study***."

2. I still do not think the section on the AIC analysis provides useful conclusions in understanding model skill. The floods being simulated are extrapolations from these distributions, and the degree to which they fit 35 years of data does not illustrate how useful they are for extrapolation.

If I am wrong about this, then please add further discussion explaining why this is a relevant test of the model, as I am presently unconvinced.

Re: Thanks, we agree with the reviewer that the degree of AIC does not necessarily mean how useful they are for extrapolation. In practice, we need to find a historical flood record with a certain reoccurrence (e.g., 1-in-100 year, 1-in-500-year) to judge if the fitting and extrapolation is good or not. Obviously, we cannot do this for the global scale. Measurement of the fitting performance with current available CaMa-Flood estimates is the only way to interpret the variations among different fitting distribution.

Moreover, the variation of the AIC will be reflected in the extrapolation. Small difference in AIC will be enlarged when extrapolated. For example, AIC for GUM always deviates from other distribution functions (Figure 3). And we can observe from Figure 6d and 7d (and other figures in the supplementary), GUM tends to provide a much higher water depth for rarer floods compared to others.

Therefore, we still think the section of AIC analysis is necessary for the analysis, and we only made slight changes in this section to include the reviewer's comments.

3. Some interpretations of the results and comparison with the literature do appear quite tokenistic, rather than contributing to a rich discussion. [a.] E.g. P18 L3-13 loosely says routing is and is not important. [b.] P18 L14-24 is a little rushed in its review of these papers and it is not clear how this relates to the study in question. P11 L5-10 suggests the spatial patterns are not consistent with Schellekens et al., but I disagree. Although the proportion of the contribution to the uncertainty by runoff forcing is fairly consistent in space (Fig 4e), the magnitude of the uncertainty (Fig 4c; the relevant figure for the point being made about Schellekens' conclusion) is larger in mountainous and arid regions.

Re: a. Thanks, we listed in the Discussion a few studies related to the river routing. However, because none of them specifically investigated the sensitivity of the flood water depth or flood inundation to the river routing, we can hardly conclude how important the river routing is. This requires further studies.

b. We have revised the Discussions. First, we combine the P18 L14-24 with the previous paragraph as both of them are discussing the potential factors affecting the flood hazard mapping. Then we add one new paragraph to discuss what are the uncertainty source can be considered in the future to the flood impact (i.e., population exposure, economic exposure). This is also a response to the reviewer's next question. Please check the text with tracks.

c. Here we just want to mention that the contribution of runoff to the uncertainties is not sensitive to the climate zones. Both Fig 3c and the figures in Schellekens et al., 2017 show that the magnitude of the uncertainties have spatial variation in mountainous and arid regions,

however, the spatial variation is not revealed in the contribution map (Figure 4e). So, we modified our texts to avoid this confusion.

> "the spatial patterns of runoff spread in their results ***and the variation in Figure 4-c*** are not seen in ***Figure 4-e***, indicating that the contribution of runoff to the total uncertainty in flood water depth is not sensitive to climate zones or topography."

4. [a.]Section 3.3 is still a fairly long-winded way of saying that runoff is most important. I would suggest condensing 3.3.1 and 3.3.2 without using so much text, as the conclusion is quite simple -- consider what the reader truly needs to know. [b.] A GRDC gauge is mentioned, but then no analysis is done using this, which is confusing. [c.]Section 3.3.3 is interesting, but the authors should be much more explicit about the uncertainties in the exposure data they are using: how accurate is 1km resolution population data? If you analysed sensitivities with the 90m flood model with higher resolution population data, I expect conclusions would be very different. There is lots of literature on the accuracy of these global population datasets and the impact of resolution that should be discussed.

Re: [a.] Thanks for the suggestions. In section 3.2, the results show that the runoff is the most important for the 1-in-100 year flood. In sections 3.3.1 and 3.3.2, we investigated if this importance is the same for all return periods. Although the results still demonstrate that runoff is the most important for all return periods, we have to use sufficient words to interpret the figures (Figure 6 & 7). Despite the conclusion for the importance of runoff, we also found the importance of the functions increases from normal return period to the rarer floods, as the uncertainty range for the rarer floods is larger than that for floods with higher reoccurrence.

We have reviewed the two sections and revised the texts but not too much. Please check the text with tracks.

[b.] The flood water depth is the variable that can be accurately measured if suffering a flood. However, as one of the reviewers mentioned in the first round reviewing, we have to select one representative location, so that we select the GRDC gauge. This will help if others conduct similar results and want to compare the results with this study.

[c.] Thanks, yes, we then found a perfect paper (Smith et al., 2019) for discussion about the resolution.

**Table 2 Total population living in the 1 in 100 year floodplain (millions) summed across all 18 countries, for varying resolutions of both hazard and population data**

|  |  | Population data | | | | | |
|  |  | 30 m | 90 m | | 900 m | | |
|  |  | HRSL | HRSL | WP | HRSL | WP | LS |
| Hazard | 90 m | 101 | 102 | 122 | 124 | 130 | 134 |
|  | 900 m | 196 | 196 | 205 | 197 | 205 | 203 |

In their paper, they evaluated the population exposure to the 1-in-100 year flood in 18 different countries but with different population products. Meanwhile they aggregated the hazard map and population data from high spatial resolution (30m or 90m) to lower spatial resolution (900m) to investigate the sensitivity of the population exposure. From their Table 2, when the hazard map is at 90m, the exposure due to resolution changes of population from 30m-90m is negligible, while increased 8-22% if population resolution decreases to 900m. When the hazard map is 900m, the exposure does not change with population resolution, while the total exposure increased by 51% to 94% compared to 90m hazard map. Therefore, the spatial resolution of the hazard map is the most import factor to the final population exposure.

In the revised manuscript, this part has been added to Discussions.

Smith, A., Bates, P. D., Wing, O., Sampson, C., Quinn, N., & Neal, J. (2019). New estimates of flood exposure in developing countries using high-resolution population data. *Nature Communications*, *10*(1), 1–7. https://doi.org/10.1038/s41467-019-09282-y

5. Some figures require improvement. The x axes on Figs 6+7 make no sense. Fig 5 is too small also (where is the yellow cross?).

Re: We have cut the x axes and only the floods corresponding to return period larger than 2 years are remained. Figure 5 has also been enlarged and the yellow cross is enlarged. Figures in the Supplementary are revised as well.